# Towards the Universal Learning Principle for Graph Neural Networks

## Abstract

Graph neural networks (GNNs) are currently highly regarded in graph repre-
sentation learning tasks due to their significant performance. Although various
propagation mechanisms and graph filters were proposed, few works have investi-
gated their rationale from the perspective of learning. In this paper, we elucidate
the criterion for the graph filter formed by power series, and further establish a
scalable regularized learning framework that theoretically realizes very deep GNN.
Following the framework, we introduce Adaptive Power GNN (APGNN), a deep
GNN that employs exponentially decaying weights to aggregate graph information
of varying orders, thus facilitating more effective mining of deeper neighbor infor-
mation. Moreover, the multiple $P$-hop message passing strategy is proposed to
efficiently perceive the higher-order neighborhoods. Different from other GNNs,
the proposed APGNN can be seamlessly extended to an infinite-depth network. To
clarify the learning guarantee, we theoretically analyze the generalization of the
proposed learning framework via uniform convergence. Experimental results show
that APGNN obtains superior performance compared to state-of-the-art GNNs,
highlighting the effectiveness of our framework.

## 1 Introduction

Recently, Graph Neural Networks (GNNs) have shown commendable performance on numerous graph
representation learning tasks. In addition, GNNs have been introduced in a variety of application tasks,
such as recommendation systems [7, 11, 37], computer vision [4, 13, 23], and traffic forecasting
[8, 9]. The fundamental part of GNN is the design of the propagation mechanism or the graph
filter [5, 12, 27, 32, 34]. GNNs can be categorized into two groups based on the approach of
formulation. Spatial-based GNN formulates propagation mechanisms through the direct aggregation
of spatial features. As one of the most simple GNNs, Graph Convolutional Network (GCN) [15]
designs graph convolutional layer via aggregating one-hop information on the graph. Graph Attention
Network (GAT) [30] learns node relationships using an attention mechanism, enhancing the scalability
of the network. For extension of inductive learning, GraphSAGE [10] employs various pooling
operations as aggregation functions. Liu et. al proposed DAGNN, which integrates information from
multiple receptive fields for adaptive propagation [19]. Spectral-based GNN designs graph filters by
constructing filter functions in the graph Fourier domain, which aims to find a proper transformation
of the graph spectrum. Chebynet constructs the localized graph filter with Chebyshev polynomial [3].
From the view of the spectrum, GCN could be seen as a Chebyshev filter with first-order truncation
[15]. To construct deeper GNN, Personalize PageRank method is employed to design graph filter
[16]. GNN-LF/HF [36] concludes various designs of graph filters and constructs the graph filter
through a graph optimization framework.

Despite their success, few studies have explored the general rule for devising GNNs from the
perspective of learning. In this paper, we start from the graph filter formed by power series and

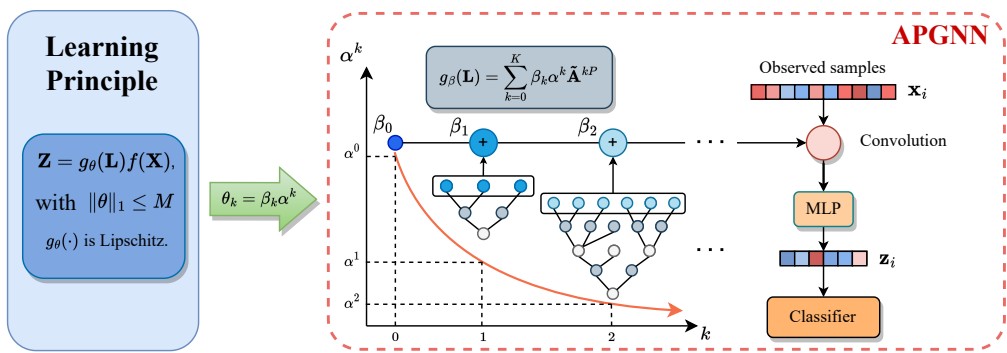

Figure 1: An illustration of the proposed APGNN that adheres to the learning principle. The model incorporates the decay rate $\alpha$ to suppress the information from high-order neighbors while adaptively learning bounded coefficients $\beta$. Furthermore, it aggregates information with $P$-hop to enlarge the receptive field. This design enables the seamless extension of APGNN to an extremely deep network.

discuss what makes a legitimate graph filter for the construction of deep GNN. A learning principle is then proposed to summarize the rule of formulating a graph filter. Following this, we propose Adaptive Power Graph Neural Network (APGNN), which adaptively learns the task-specific graph filter for node representation learning. The main idea of APGNN is depicted in Figure 1. The parameterized graph filter is designed with regularization of the exponential decay rate. A multiple $P$-hop strategy is applied to enhance the capacity of perceiving the higher-order neighborhoods. Furthermore, the generalization bound of APGNN is presented with the setting of the continuous graph, which provides a learning guarantee for the proposed principle theoretically.

We conduct evaluations on five benchmark datasets on node classification tasks. The experimental results suggest the superiority of the proposed method over the existing GNNs. The theoretical analysis is also validated via the empirical study.

## 2 Preliminaries

**Notations.** Suppose we have an undirected graph $\mathcal{G} = (\mathcal{V}, \mathcal{E}, \mathbf{A})$ with node set $\mathcal{V}$ and $|\mathcal{V}| = n$. $\mathbf{A} \in \mathbb{R}^{n \times n}$ denotes the adjacency matrix indicating the edges in $\mathcal{E}$. Assuming that the self-loops are contained in the graph, i.e., $a_{ii} = 1$. Let $\mathbf{X} = [\mathbf{x}_1, \mathbf{x}_2, \cdots, \mathbf{x}_n]^\top \in \mathbb{R}^{n \times d}$ be the graph signals (or features) of the nodes. We use notation $[n] \triangleq \{1, 2, \cdots, n\}$ for $n \in \mathbb{N}_+$. Assume that the label of $\mathbf{x}_i$ is $y_i \in \mathcal{Y}$ for all $i = [n_l]$, where $n_l \leq n$ is the number of labeled samples.

**Graph Neural Networks.** We introduce some essential concepts in GNNs. Let $d_i = \sum_{j=1}^{n} A_{ij}$ be the degree of $i$-th node, so the degree matrix of $\mathbf{A}$ can be defined as $\mathbf{D} = \mathrm{diag}(d_1, d_2, \cdots, d_n)$. The symmetrically normalized Laplacian is $\mathbf{L} = \mathbf{I} - \tilde{\mathbf{A}}$, where $\tilde{\mathbf{A}} \triangleq \mathbf{D}^{-1/2}\mathbf{A}\mathbf{D}^{-1/2}$ is normalized adjacency matrix. Consider the eigen-decomposition $\mathbf{L} = \mathbf{U}\mathbf{\Lambda}\mathbf{U}^\top$, where $\mathbf{\Lambda} = \mathrm{diag}(\lambda_1, \cdots, \lambda_n)$ is the diagonal matrix of eigenvalues, and $\mathbf{U} = [\mathbf{u}_1, \cdots, \mathbf{u}_n]$ represents the eigenvectors associated with the eigenvalues. Note that $\tilde{\mathbf{A}}$ shares the same eigenvectors with $\mathbf{L}$.

Spectral convolution on graphs is defined as the following transformation [15, 28]:

$$g * \mathbf{X} = \mathbf{U}g(\mathbf{\Lambda})\mathbf{U}^\top\mathbf{X}, \tag{1}$$

where $g(\cdot) : [0, 2] \mapsto \mathbb{R}$ is called filter function and $g(\mathbf{\Lambda}) = \mathrm{diag}(g(\lambda_1), \cdots, g(\lambda_n))$. The common approach in GNNs is to apply polynomial functions as the filters [3, 12, 15], which leads to $\mathbf{U}g(\mathbf{\Lambda})\mathbf{U}^\top = g(\mathbf{L})$. Therefore, spectral convolution is usually written as $g * \mathbf{X} = g(\mathbf{L})\mathbf{X}$. The graph representation paradigm in GNN is generally expressed as follows:

$$\mathbf{Z} = g(\mathbf{L})f(\mathbf{X}), \quad g(\mathbf{L}) = \sum_{k=0}^{K} \theta_k \tilde{\mathbf{A}}^k, \tag{2}$$

where $\mathbf{Z} \in \mathbb{R}^{n \times c}$ denotes the node representation, and $f(\cdot)$ represents a feature extractor such as multi-layer perceptions (MLPs).

## 3 Learning Principle for GNNs

### 3.1 The principle of devising graph filters

Current studies suggest a significant relationship between the performance of GNN and its graph filter [16, 19]. Predominantly, the general graph filters are characterized by polynomials associated with the adjacency matrix $\tilde{\mathbf{A}}$ (or Laplacian matrix $\mathbf{L}$), i.e., $g(\mathbf{L}) = \sum_{k=0}^{K} \theta_k \tilde{\mathbf{A}}^k$. However, the existing methods still meet the issue that the depth of GNN is limited. The reason for this phenomenon is that these GNNs are inconsistent with their "infinite-depth" version. That is, the corresponding models lose some essential properties as the depth $K \to \infty$. Consequently, the depth of the models is restricted. To address this issue, it is necessary to study the properties of GNNs with infinite depth. Therefore, we explore the graph filter reformulated as power series:

$$g(\tilde{\mathbf{A}}) = \sum_{k=0}^{\infty} \theta_k \tilde{\mathbf{A}}^k = \sum_{k=0}^{\infty} \theta_k (\mathbf{I} - \mathbf{L})^k. \tag{3}$$

First of all, a well-defined graph filter represented as equation 3 must be convergent. Consequently, it becomes essential to investigate the properties that the coefficients $\theta_k$ should exhibit. The following lemma provides appropriate constraints for the coefficients of the graph filter.

**Lemma 1.** *Let $\{a_k\}$ and $\{\gamma^k\}$ be the real number sequences, where $\gamma \in (-1, 1]$ and $k \in \mathbb{N}$. Then $\sum_{k}^{\infty} a_k \gamma^k$ converges uniformly and absolutely if and only if the series $\sum_{k}^{\infty} a_k$ converges absolutely.*

As a direct corollary, the weights of the graph filter (i.e., $\theta_k$) should satisfy the following theorem.

**Theorem 1.** *Let $\tilde{\mathbf{A}} = \mathbf{D}^{-1/2} \mathbf{A} \mathbf{D}^{-1/2}$ be the normalized adjacency matrix of a graph $\mathcal{G}$ with spectral radius $\rho(\tilde{\mathbf{A}}) \leq 1$. The matrix series $\sum_{k=0}^{\infty} \theta_k \tilde{\mathbf{A}}^k$ converges uniformly and absolutely if and only if the series $\sum_{k=0}^{\infty} \theta_k$ converges absolutely.*

The proofs are shown in Appendix. Theorem 1 offers a sufficient and necessary condition for the convergence of graph filters formed by power series. Specifically, the condition requires the existence of a finite real number $M \geq 0$,

$$\|\boldsymbol{\theta}\|_1 \triangleq \sum_{k=0}^{\infty} |\theta_k| \leq M. \tag{4}$$

Therefore, an arbitrary graph filter formed by power series should satisfy the above convergence condition, which gives the first requirement while designing GNN. Apart from convergence, we expect the graph filter to possess good analytic properties such as smoothness. To this end, Lipschitz continuity should be considered the second requirement of the graph filter. Let $g(\cdot)$ be an $L$-Lipschtiz continuous function, meaning that

$$|g(\lambda) - g(\lambda')| \leq L|\lambda - \lambda'|, \quad \forall \lambda, \lambda' \in [0, 2). \tag{5}$$

This property indicates the stability or robustness of the model [6, 24]. If the graph is contaminated and its eigenvalues are perturbed by at most $\epsilon$, Lipschitz continuity ensures the perturbation of the graph-filtered result is at most $L\epsilon$. For instance, considering $g(\lambda) = \sum_{k=0}^{\infty} (1 - \lambda)^k / k^2$, which is convergent, yet the Lipschitz condition does not satisfy for $\lambda$ closed to zero. Therefore, this graph filter might be sensitive to the input graph. Subsequently, we conclude the following criterion.

$$\mathbf{Z} = g_{\boldsymbol{\theta}}(\mathbf{L}) f(\mathbf{X}), \text{ with } \|\boldsymbol{\theta}\|_1 \leq M, \ g_{\boldsymbol{\theta}}(\cdot) \text{ is a Lipschitz function.} \tag{6}$$

To enhance the scalability of the model, we define $\boldsymbol{\theta}$ as a learnable parameter (though its dimension is infinite). In this way, (6) gives a regularized learning framework for GNN. Therefore, for a $K$-order polynomial graph filter $g_{\boldsymbol{\theta}}^K(\lambda) = \sum_{k=0}^{K} \theta_k (1 - \lambda)^k$, which is what we can implement in practice, the condition (6) should be satisfied to keep the consistency with its infinitely deep version $g_{\boldsymbol{\theta}}^{\infty}(\lambda) = \sum_{k=0}^{\infty} \theta_k (1 - \lambda)^k$. We will present the applications of this criterion in this section, and further analyze the learning guarantee with generalization in section 4.

### 3.2 Related works

In this subsection, we investigate the relationship between our learning framework and several well-known Graph Neural Networks (GNNs), focusing on the design of graph filters. Our findings indicate that these GNNs are all special cases of our learning framework, which are summarized in Table 1.

**GCN/SGC [15, 33].** Graph convolutional network (GCN) aims to learn a node representation by stacking multiple graph convolutional layers. In each layer, GCN applies first-order Chebyshev approximation as the graph filter followed by a fully connected layer. For simplicity, we analyze one-layer GCN, which is formulated as $\mathbf{Z} = \sigma(\tilde{\mathbf{A}}\mathbf{X}\mathbf{W})$, where $\mathbf{W}$ is a learnable weight matrix for linear transformation. Therefore, the graph filter of one-layer GCN is $g_{\text{GCN}}(L) = \mathbf{I} - \mathbf{L} = \tilde{\mathbf{A}}$, or a trivial matrix power series:

$$g_{\text{GCN}}(\mathbf{L}) = \sum_{k=0}^{\infty} \theta_k \tilde{\mathbf{A}}^k, \quad \text{where } \theta_k = \begin{cases} 1, & \text{if } k = 1, \\ 0, & \text{otherwise.} \end{cases} \tag{7}$$

It should be noted that this equation satisfies the condition described in (6).

SGC is a simplified version of GCN that eliminates the activation function and applies a single linear projection to extract features. This simplification reduces the multiple-layer GCN into a more concise model as $\mathbf{Z} = \tilde{\mathbf{A}}^K \mathbf{X}\mathbf{W}$. Similarly, the graph filter of SGC can be represented as:

$$g_{\text{SGC}}(\mathbf{L}) = \tilde{\mathbf{A}}^K = \sum_{k=0}^{\infty} \theta_k \tilde{\mathbf{A}}^k, \quad \text{where } \theta_k = \begin{cases} 1, & \text{if } k = K, \\ 0, & \text{otherwise.} \end{cases} \tag{8}$$

Both GCN and SGC use a monomial to construct the graph filter. Therefore, in the viewpoint of spectral-GNN, their graph filters are too simple to capture the spectral characteristic. Besides, the small eigenvalue vanishes when $K$ becomes very large, leaving only the largest eigenvalue, which leads to the well-known over-smoothing problem [17].

**PPNP [16].** PPNP uses Personalized PageRank as the graph filter, which balances local information preservation and the utilization of high-order neighbor information. The model of PPNP is $\mathbf{Z} = \alpha(\mathbf{I} - (1-\alpha)\tilde{\mathbf{A}})^{-1}\mathbf{H} = (\mathbf{I} + \beta\mathbf{L})^{-1}\mathbf{H}$, where $\mathbf{H} = f(\mathbf{X})$ is a two-layer MLPs and $\beta = 1/\alpha - 1$. Hence, the graph filter of PPNP is $g_{\text{PPNP}}(\mathbf{L}) = (\mathbf{I} + \beta\mathbf{L})^{-1}$. Considering its Taylor series, we have

$$g_{\text{PPNP}}(\mathbf{L}) = (\mathbf{I} + \beta\mathbf{L})^{-1} = \frac{1}{1+\beta} \sum_{k=0}^{\infty} \left( \frac{\beta}{1+\beta} \right)^k \tilde{\mathbf{A}}^k = \sum_{k=0}^{\infty} \theta_k \tilde{\mathbf{A}}, \tag{9}$$

where $\theta_k = \beta^k/(1+\beta)^{k+1}$. It is straightforward to validate that $\sum_{k=0}^{\infty} \theta_k = 1$, and thus the convergence requirement (4) holds. Moreover, the Lipschitz condition is easily verified. Thus PPNP satisfies the criterion of (6). However, the performance of PPNP is heavily dependent on the hyperparameter $\beta$, which must be carefully tuned to achieve optimal performance.

**DAGNN [19].** DAGNN adaptively adjusts the weight of information aggregation from different neighbors to solve the over-smoothing problem. It designs a parameterized graph filter formulated as a $K$-order polynomial:

$$g_{\text{DAGNN}}(\mathbf{L}) = \sum_{k=0}^{K} \theta_k \tilde{\mathbf{A}}^k, \quad \text{s.t. } 0 \leq \theta_k \leq 1, \tag{10}$$

where $\theta_k$ is the learnable parameter with bounded constraint. Due to this adaptive learning strategy, DAGGN is able to learn a graph filter more suitable for node classification. The empirical studies suggest DAGNN works well with a proper $K$. However, as $K \to \infty$, the constraint $0 \leq \theta_k \leq 1$ cannot guarantee the convergence of the graph filter. It indicates that DAGNN is "inconsistent" with its infinitely deep version. Therefore, it can not be naturally extended to significantly deep GNN.

### 3.3 Instantiation: Adaptive Power Graph Neural Network

We now introduce a novel GNN following the framework in section 3.1, called Adaptive Power GNN (APGNN). We first consider the following graph filter parameterized by $\boldsymbol{\beta}$ with the form:

$$g_{\boldsymbol{\beta}}^{\infty}(\lambda) = \sum_{k=0}^{\infty} \beta_k \alpha^k (1 - \lambda)^k, \quad \text{where } |\beta_k| \leq 1, \ 0 < \alpha < 1, \tag{11}$$

where the coefficient of the power series $\theta_k = \beta_k \alpha^k$, with hyper-parameter $\alpha \in (0, 1)$ ensuring the convergence. Immediately, we check the condition of Lemma 1.

$$\|\boldsymbol{\theta}\|_1 = \sum_{k=0}^{\infty} |\beta_k \alpha^k| \leq \sum_{k=0}^{\infty} \alpha^k \leq \frac{1}{1-\alpha}. \tag{12}$$

Table 1: Graph filter for various GNNs

| Model | Filter function | Setting of $\boldsymbol{\theta}$ | Learnable $g(\cdot)$ |
|---|---|---|---|
| 1-layer GCN | $g(\mathbf{L}) = \sum_{k=0}^{\infty} \theta_k \tilde{\mathbf{A}}^k$ | $\theta_k = \begin{cases} 1, & \text{if } k = 1 \\ 0, & \text{otherwise} \end{cases}$ | No |
| SGC | $g(\mathbf{L}) = \sum_{k=0}^{\infty} \theta_k \tilde{\mathbf{A}}^k$ | $\theta_k = \begin{cases} 1, & \text{if } k = K \\ 0, & \text{otherwise} \end{cases}$ | No |
| PPNP | $g(\mathbf{L}) = \sum_{k=0}^{\infty} \theta_k \tilde{\mathbf{A}}^k$ | $\theta_k = \dfrac{\beta^k}{(1+\beta)^{k+1}}, \ \beta > 0$ | No |
| DAGNN | $g(\mathbf{L}) = \sum_{k=0}^{K} \theta_k \tilde{\mathbf{A}}^k$ | $0 \le \theta_k \le 1$ | Yes |

Hence, the power series converges on $[0, 2]$ absolutely and uniformly. Similarly, the associated matrix series $g_{\boldsymbol{\beta}}^{\infty}(\mathbf{L}) = \sum_{k=0}^{\infty} \beta_k \alpha^k \tilde{\mathbf{A}}^k$ also converges uniformly and absolutely by Theorem 1. Moreover, $g_{\boldsymbol{\beta}}^{\infty}(\cdot)$ is $\alpha(1-\alpha)^{-2}$-Lipschitz. To see this, for any $|\beta_k| \le 1$ and $1 - \lambda \in (-1, 1]$, we have

$$|\nabla g_{\boldsymbol{\beta}}^{\infty}(\lambda)| = \left| \sum_{k=1}^{\infty} (-1)^k k \beta_k \alpha^k (1-\lambda)^{k-1} \right| \le \sum_{k=1}^{\infty} k \alpha^k = \frac{\alpha}{(1-\alpha)^2}, \tag{13}$$

which implies the Lipschitz continuous property. Thus, this graph filter fits the requirement of the proposed criterion. However, the model with this graph filter is unavailable in practice as the number of parameters to be learned is infinite. The $K$-order truncated polynomial is utilized for substitution, i.e., $g_{\boldsymbol{\beta}}^{K}(\mathbf{L}) = \sum_{k=0}^{K} \beta_k \alpha^k \tilde{\mathbf{A}}^k$. We evaluate the approximation via the upper bound of $K$-order truncation error:

$$|g_{\boldsymbol{\beta}}^{\infty}(\lambda) - g_{\boldsymbol{\beta}}^{K}(\lambda)| \le \sum_{k=K+1}^{\infty} |\beta_k \alpha^k (1-\lambda)^k| \le \sum_{k=K+1}^{\infty} \alpha^k = \frac{\alpha^{K+1}}{1-\alpha}, \tag{14}$$

which uniformly holds for $\forall \lambda \in [0, 2]$. Likewise, the approximation error of matrix series is given by

$$\left\| g_{\boldsymbol{\beta}}^{\infty}(\mathbf{L}) - g_{\boldsymbol{\beta}}^{K}(\mathbf{L}) \right\|_2 = \left\| \mathbf{U} \left( g_{\boldsymbol{\beta}}^{\infty}(\boldsymbol{\Lambda}) - g_{\boldsymbol{\beta}}^{K}(\boldsymbol{\Lambda}) \right) \mathbf{U}^{\top} \right\|_2 = \sup_{i \in [n]} |g_{\boldsymbol{\beta}}^{\infty}(\lambda_i) - g_{\boldsymbol{\beta}}^{K}(\lambda_i)| \le \frac{\alpha^{K+1}}{1-\alpha}, \tag{15}$$

where $\lambda_i$ denotes the $i$-th eigenvalue of $\mathbf{L}$. This upper bound is independent of the given graph, which can be controlled via tuning $\alpha$ and $K$. The higher $K$ and smaller $\alpha$ yield a better approximation to the exact graph filter $g_{\boldsymbol{\beta}}^{\infty}(\cdot)$. Nevertheless, the small $\alpha$ tends to limit the capability of the graph filter. Extremely, $\alpha \to 0$ gives a trivial function $g_{\boldsymbol{\beta}}^{K}(\lambda) = \beta_0$. This suggests that $\alpha$ should be elaborately tuned to improve the performance.

Though the aforementioned graph filter is primarily motivated via spectral analysis, we can still present the spatial perspective explanation for its design. Existing GNNs aggregate the neighbor information of different hops with certain weights, which could be either manually assigned or learned adaptively. Typically, methods like GPR-GNN [2] and DAGNN [19] that learn the aggregation weight, tend to treat the neighbor's information of different hops equally. That is, the $k$-hop's weight are assigned with $\theta_k = \mathcal{O}(1)$ for each $k \in [K]$. However, it is shown in the previous research that the propagation with the very high-order neighbor potentially leads to the over-smoothing issue [25, 33]. The current methods magnify this flaw of the high-order graph since they cannot distinguish the significance of the information of different hops. This motivates the design of the decay rate in APGNN, i.e., we employ weights with exponential decaying rate by assigning $\theta_k = \mathcal{O}(\alpha^k)$ for some $0 < \alpha < 1$. This approach emphasizes the contribution of lower-order neighbors and restricts the over-weighting of the information from high-order neighbors due to $\theta_k \to 0$ with $k \to \infty$. Therefore, it provides more effective aggregation and thus enhances the model's scalability.

To take a further step in the construction of a deep GNN, we introduce a multiple $P$-hop strategy for the graph filter of (11), which effectively extends the utmost neighborhood range that the graph filter can perceive by $P$ times. Consider a different perspective regarding the construction of a filter with

the utmost order $T = KP$. The previous methods can be viewed as a one-hop graph filter by setting $P = 1$. For $P > 1$, the graph filter is able to aggregate information from a larger neighborhood in the same order. In addition, we will illustrate the advantages of this strategy from the perspective of generalization in the following section.

Summarizing the above analysis, we present the following comprehensive architecture of APGNN:

$$\mathbf{Z} = g_{\boldsymbol{\beta}}(\mathbf{L})f(\mathbf{X}), \quad f(\mathbf{X}) = \mathrm{MLP}(\mathbf{X}), \quad g_{\boldsymbol{\beta}}(\mathbf{L}) = \sum_{k=0}^{K} \beta_k \alpha^k \tilde{\mathbf{A}}^{kP}. \tag{16}$$

In short, APGNN incorporates the benefits from the decay rate $\alpha$ that exponentially suppresses the information of extremely high-order neighbors and the multiple $P$-hop strategy to enlarge receptive fields. These approaches make it possible to realize a sufficiently deep GNN.

## 4 Generalization analysis

The theoretical analysis of GNN's generalization is widely studied. [31] provides the generalization result of the algorithmic stability of GCN in the discrete graph setting. In contrast, [14] shows the convergence and stability guarantee over the random and continuous graph. In this section, we will present the uniform generalization bound of the proposed GNN learning framework under the continuous setup.

We first introduce some notations for later discussion. Denote $\mathbf{x} \in \mathcal{X}$ as any samples from the input space $\mathcal{X}$ (we generally set $\mathcal{X}$ as a subset of $\mathbb{R}^d$). Let $\rho(\cdot)$ be a probability measure defined over $\mathcal{X}$. Assume $x_j$ is the $j$-th coordinate of $\mathbf{x} \in \mathcal{X}$ and $\mathbb{E}[x_j^2] \leq c_{\mathcal{X}}^2$ for any $j \in [d]$. To describe the graph relation between each pair $(\mathbf{x}, \mathbf{x}')$ over $\mathcal{X} \times \mathcal{X}$, we define a continuous graph function $A(\cdot, \cdot) : \mathcal{X} \times \mathcal{X} \mapsto \mathbb{R}_+$, and its corresponding degree function is

$$d(\mathbf{x}') = \int_{\mathcal{X}} A(\mathbf{x}, \mathbf{x}')\mathrm{d}\rho(\mathbf{x}'). \tag{17}$$

Different from the setting of [18, 26], we assume $0 \leq A(\mathbf{x}, \mathbf{x}') \leq c_U$, and $0 < c_L \leq d(\mathbf{x})$ for any $\mathbf{x}, \mathbf{x}' \in \mathcal{X}$. Therefore, we can define the symmetric normalized graph:

$$\tilde{A}(\mathbf{x}, \mathbf{x}') = \frac{A(\mathbf{x}, \mathbf{x}')}{\sqrt{d(\mathbf{x})d(\mathbf{x}')}}. \tag{18}$$

Then the corresponding normalized Laplacian is $L = I - \tilde{A}$, where $I$ indicates the identity operator over $\mathcal{X}$. For a graph filter function $g_{\boldsymbol{\theta}}(\lambda) = \sum_{k=0}^{K} \theta_k (1 - \lambda)^k$, graph convolution of the continuous graph is defined as the following integral operator:

$$g_{\boldsymbol{\theta}} L f = \sum_{k=0}^{K} \theta_k \tilde{A}^k f, \quad \tilde{A}f = \int_{\mathcal{X}} \tilde{A}(\cdot, \mathbf{x}) f(\mathbf{x})\mathrm{d}\rho(\mathbf{x}), \tag{19}$$

where $\tilde{A}^k = \tilde{A}^{k-1} \circ \tilde{A}$ denotes $k$-order composition of integral operator with $\tilde{A}^0 = I$. Note we have $\sum_{k=0}^{K} \theta_k \|\tilde{A}\| \leq \|\boldsymbol{\theta}\|_1 \leq M$ for any $K \in \mathbb{N}$, indicating $\sum_{k=0}^{\infty} \theta_k \tilde{A}$ is absolutely summable. This guarantees the existence of graph filter on the continuous graph when $K \to \infty$. For convenience in understanding, we provide the analysis on a simplified GNN, where we consider a semi-supervised learning task with two classes, i.e., $y_i \in \mathcal{Y} \triangleq \{-1, 1\}$, and utilize linear feature extractor $f(\mathbf{X}) = \mathbf{w}^{\top}\mathbf{X}$. Note that we can still extend our result for $f(\mathbf{X}) = \mathrm{MLP}(\mathbf{X})$ and multi-class cases using the techniques proposed in [1]. With the above setting, the hypothesis set over is described as

$$\mathcal{H}_{\mathcal{X}} = \{h : h(\mathbf{x}) = g_{\boldsymbol{\theta}} L f(\mathbf{x}), \ f(\mathbf{x}) = \langle \mathbf{w}, \mathbf{x} \rangle, \ \|\mathbf{w}\|_2 \leq B, \ \|\boldsymbol{\theta}\|_1 \leq M\}. \tag{20}$$

However, the integral in each hypothesis $h \in \mathcal{H}_{\mathcal{X}}$ is intractable since the underlying graph function and the data distribution are unknown. Therefore, we should use the "empirical version" of the hypothesis to estimate $h \in \mathcal{H}_{\mathcal{X}}$. For this reason, we introduce the hypothesis set defined over the observed samples $S$ and graph $\mathcal{G}$:

$$\mathcal{H}_S = \left\{ h : h(\mathbf{x}_i) = \sum_{j=1}^{n} g_{\boldsymbol{\theta}}(\mathbf{L})_{ij} \mathbf{x}_j^{\top} \mathbf{w}, \quad \|\mathbf{w}\|_2 \leq B, \ \|\boldsymbol{\theta}\|_1 \leq M \right\}. \tag{21}$$

Define the generalization error and the empirical error [21] as follows

$$R(h) = \mathbb{E}_{(\mathbf{x},y)}[1_{yh(\mathbf{x}) \leq 0}], \quad \hat{R}(h) = \frac{1}{n_l} \sum_{i=1}^{n_l} \min(1, \max(0, 1 - y_i h(\mathbf{x}_i))). \quad (22)$$

We have the following theorem on the generalization of the proposed learning paradigm.

**Theorem 2.** *Suppose $g_{\boldsymbol{\theta}}(\cdot)$ is $L_M$-Lipschitz. Let $h_{\mathbf{w},\boldsymbol{\theta}} \in \mathcal{H}_{\mathcal{X}}$ and $h_{\mathbf{w},\boldsymbol{\theta}} \in \mathcal{H}_S$ share the same parameter $(\mathbf{w}, \boldsymbol{\theta})$. Then there exists a constant $C > 0$ related to the graph function, with the probability at least $1 - \delta$, the following inequality holds.*

$$R(h_{\mathbf{w},\boldsymbol{\theta}}) \lesssim \hat{R}(\hat{h}_{\mathbf{w},\boldsymbol{\theta}}) + 2BMc_{\mathcal{X}} \sqrt{\frac{2d \log(2K + 2)}{n_l}} + BCL_M dc_{\mathcal{X}} \sqrt{\frac{\log(2/\delta)}{n}}. \quad (23)$$

The proof is given by excess risk decomposition, shown in Appendix. The notation "$\lesssim$" denotes "less than or approximately equal to the right-hand side" and guarantees an approximation error of at most $\mathcal{O}(\sqrt{\frac{\log(1/\tau)}{n_l}})$ with a probability of at least $1 - \mathcal{O}(\tau)$. We remind readers the important difference between $R(h_{\mathbf{w},\boldsymbol{\theta}})$ and $\hat{R}(\hat{h}_{\mathbf{w},\boldsymbol{\theta}})$. The former term measures the population error over the whole input space with the **continuous** graph filter $g_{\boldsymbol{\theta}} L$. In contrast, $\hat{R}(\hat{h}_{\mathbf{w},\boldsymbol{\theta}})$ is the empirical risk (i.e., training risk) on the sample set $S$ with the **discrete** graph filter $g_{\boldsymbol{\theta}}(\mathbf{L})$. $h_{\mathbf{w},\boldsymbol{\theta}}$ shares the same learning parameter with $\hat{h}_{\mathbf{w},\boldsymbol{\theta}}$. Therefore, the minimization of the right-hand-side of (23) w.r.t $(\mathbf{w}, \boldsymbol{\theta})$ reduces the upper bound of the population error.

We observe the first term of generalization bound is of order $\mathcal{O}((dn_l^{-1} \log K)^{1/2})$, which outlines the model's complexity. Although it becomes infinity when $K \to \infty$, the growth of this term is extremely slow as $K$ increases. In practice, we generally set $K < n$ since the neighbor information beyond $n$-hops is redundant, restricting the complexity away from infinity. Therefore, the generalization of the model is rigorously guaranteed for sufficiently large $K$, which allows us to construct significantly deep GNN in the proposed framework. We can obtain a more precise estimation for a certain model. In the following proposition, we unveil the generalization of APGNN as a direct application of Theorem 2.

**Proposition 1.** *Let $\boldsymbol{\beta} \in \mathbb{R}^K$ and $g_{\boldsymbol{\beta}}^K(\lambda) = \sum_{k=0}^{K} \beta_k \alpha^k (1 - \lambda)^k$ where $0 < \alpha < 1$ and $\|\boldsymbol{\beta}\|_{\infty} \leq 1$. with the probability at least $1 - \delta$, the following inequality holds.*

$$R(h_{\mathbf{w},\boldsymbol{\beta}}) \lesssim \hat{R}(\hat{h}_{\mathbf{w},\boldsymbol{\beta}}) + \frac{2Bc_{\mathcal{X}}(1 - \alpha^K)}{1 - \alpha} \sqrt{\frac{2d \log(2K + 2)}{n_l}} + \frac{BCdc_{\mathcal{X}} \alpha}{(1 - \alpha)^2} \sqrt{\frac{\log(2/\delta)}{n}}. \quad (24)$$

*Proof.* This is a direct result with $M = (1 - \alpha^K)/(1 - \alpha)$ and $L_M = \alpha/(1 - \alpha)^2$ in Theorem 2. $\square$

In (24), the complexity term becomes $\mathcal{O}(\sqrt{\log K}(1 - \alpha^K))$ with $K = \lfloor T/P \rfloor$, which is relatively tighter than $\mathcal{O}(\sqrt{\log K})$. For this term, we promote further discussion with $P$-hop. Since it takes $\lfloor T/P \rfloor$ steps to reach the $T$-order graph, the term becomes $\mathcal{O}(\sqrt{\log \lfloor T/P \rfloor}(1 - \alpha^{\lfloor T/P \rfloor}))$. It is observed that the term decreases as $P$ increases. Therefore, the appropriate $P$ reduces the bound, explaining the mechanism of the $P$-hop method. On the other hand, larger $\alpha$ leads to a higher bound. From the point of spatial view, the information from high-order neighbors is underused, which limits the range of the graph filter. Thus $\alpha$ should be moderate to leverage the generalization and the capability of the model.

## 5 Experiment

In this section, we conduct node classification experiments on various benchmark datasets to evaluate the performance of APGNN. Specifically, we compare our method with state-of-the-art methods and display the corresponding learned graph filter on different data sets. Moreover, to validate the theoretical analysis, the influence of parameters $K$, $\alpha$, and $P$ is also investigated in experiments.

Table 2: The average accuracy (%) and standard deviation (%) on five benchmark datasets. The highest accuracy in each column is shown in bold, while the second-best result is underlined.

| Model | Dataset | | | | |
|---|---|---|---|---|---|
| | Cora | Citeseer | Pubmed | Wiki-CS | MS-Academic |
| MLP | $57.79_{\pm 0.11}$ | $61.20_{\pm 0.08}$ | $73.23_{\pm 0.05}$ | $65.66_{\pm 0.20}$ | $87.79_{\pm 0.42}$ |
| ChebNet | $79.92_{\pm 0.18}$ | $70.90_{\pm 0.37}$ | $76.98_{\pm 0.16}$ | $63.24_{\pm 1.43}$ | $90.76_{\pm 0.73}$ |
| GCN | $82.03_{\pm 0.27}$ | $71.05_{\pm 0.33}$ | $79.26_{\pm 0.18}$ | $72.05_{\pm 0.45}$ | $92.07_{\pm 0.13}$ |
| SGC | $81.89_{\pm 0.26}$ | $\underline{72.18}_{\pm 0.24}$ | $78.58_{\pm 0.15}$ | $72.76_{\pm 0.35}$ | $89.01_{\pm 0.40}$ |
| GAT | $82.82_{\pm 0.36}$ | $71.96_{\pm 0.39}$ | $79.15_{\pm 0.34}$ | $74.36_{\pm 0.58}$ | $91.86_{\pm 0.27}$ |
| GraphSage | $82.14_{\pm 0.25}$ | $71.80_{\pm 0.36}$ | $79.20_{\pm 0.27}$ | $73.17_{\pm 0.41}$ | $91.53_{\pm 0.15}$ |
| PPNP | $83.73_{\pm 0.31}$ | $71.74_{\pm 0.44}$ | $80.28_{\pm 0.22}$ | $74.69_{\pm 0.53}$ | $92.58_{\pm 0.06}$ |
| APPNP | $83.73_{\pm 0.21}$ | $71.70_{\pm 0.21}$ | $80.07_{\pm 0.21}$ | $74.91_{\pm 0.61}$ | $92.81_{\pm 0.12}$ |
| GNN-LF(iter) | $\underline{83.83}_{\pm 0.36}$ | $71.44_{\pm 0.42}$ | $\underline{80.31}_{\pm 0.16}$ | $75.19_{\pm 0.49}$ | $92.78_{\pm 0.22}$ |
| GNN-HF(iter) | $83.68_{\pm 0.31}$ | $71.58_{\pm 0.36}$ | $79.99_{\pm 0.22}$ | $74.71_{\pm 0.55}$ | $92.72_{\pm 0.31}$ |
| DAGNN | $82.70_{\pm 0.17}$ | $71.90_{\pm 0.06}$ | $80.06_{\pm 0.30}$ | $\underline{75.63}_{\pm 0.48}$ | $\underline{93.24}_{\pm 0.21}$ |
| Ours | $\mathbf{84.15}_{\pm 0.23}$ | $\mathbf{72.44}_{\pm 0.56}$ | $\mathbf{80.74}_{\pm 0.24}$ | $\mathbf{76.03}_{\pm 0.51}$ | $\mathbf{93.69}_{\pm 0.20}$ |

## 5.1 Experiment Setup

**Datasets.** We perform experiments on five benchmark datasets commonly used in node classification tasks. **1). Cora, Citeseer, Pubmed[29, 35]**: These are three standard citation networks where each node is a paper and each edge is a citation link. **2).Wiki-CS[20]**: This dataset defines the computer science articles as nodes, while the hyperlinks are edges. **3). MS Acadamic[16]**: The nodes represent the author and the edges represent the co-authorships. A co-authorship Microsoft Academic Graph, where the nodes are the bag-of-words representation of the papers' abstract and edges are co-authorship. The data statistics and their partitions are presented in Appendix.

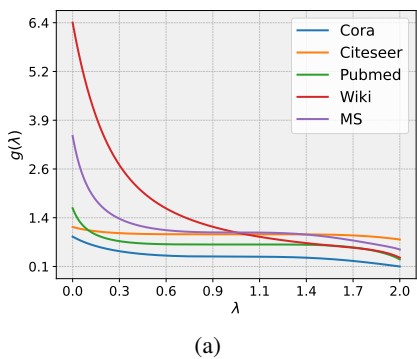
(a)

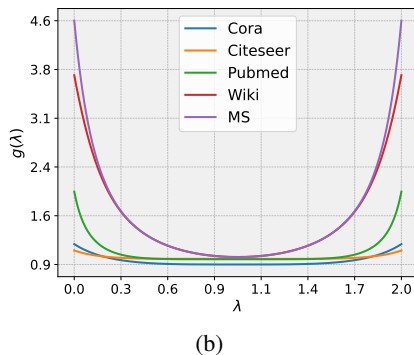
(b)

Figure 2: The graph filters learned on different data sets, with the parameter $P$ being odd in subfigure (a) and even in subfigure (b).

**Baselines.** To evaluate the effectiveness of APGNN, we compare it with the following baseline models: 1) MLP [22], a traditional method that does not use graphs, 2) GAT [30] and GraphSAGE [10], spatial methods that aggregate neighborhoods' information, and 3) ChebNet [3], GCN [15], SGC [33], PPNP, APPNP[16], GNN-LF (iteration form), GNN-HF (iteration form) [36], and DAGNN [19], spectral methods analyzing GNNs with graph Fourier transform.

**Settings.** We conducted 10 runs for each method on each dataset, with a hidden dimension of $64$. For all compared methods, their parameter settings follow the previous practices [19, 36]: the dropout rate is $0.5$ except for Cora, which had a rate of $0.8$. Furthermore, the learning rate is $0.01$ for Cora, Citeseer, and Pubmed, but $0.03$ for Wiki-CS and $0.02$ for MS-Academic, while the weight decay is $0.005$ for Cora and Pubmed, $0.02$ for Citeseer, $0.0005$ for Wiki-CS, and $0.00525$ for MS-Academic. We fix the polynomial order $K$ to 10 in ChebNet, APPNP, GNN-LF, GNN-HF, DAGNN, and APPNP. The best hyperparameters we choose for APGNN are presented in Appendix. To ensure a fair

comparison with the compared methods, we also applied our optimal hyperparameters to them, selecting the maximum value to display.

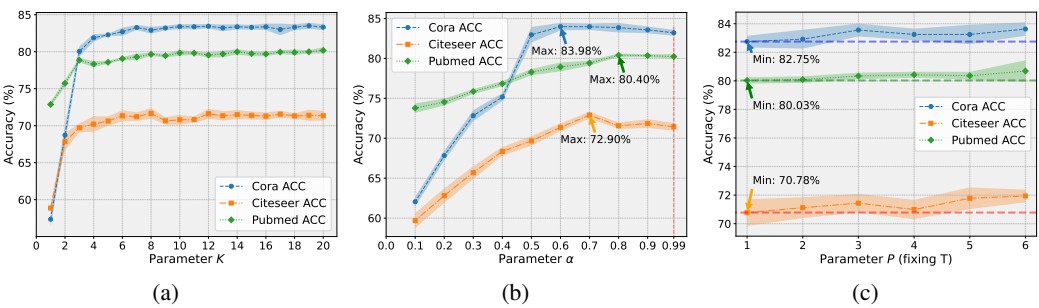

Figure 3: Accuracy with different (a) $K$. (b) $\alpha$. (c) $P$ (for fixing $T$).

## 5.2 Analysis

**Node Classification.** As the metric for evaluation, the mean accuracy of 10 runs is used. We compare the performance of APGNN with other methods on five benchmark datasets. Experiment results are reported in Table 2. We can observe that APGNN achieves the highest accuracy across all five datasets, demonstrating its superior performance.

**Learnable Graph Filters.** Figure 2 shows the graph filters learned on various datasets via APGNN. When the parity of $P$ varies, the graph filter has a distinctive shape. However, their shapes exhibit minimal impact on their accuracy regardless of the parity of $P$ according to the experiment results. Moreover, the graph filters of each dataset are plotted in Appendix, more details are included in Appendix. Our results show that the graph filters learned from different datasets vary in detail, even when their parameters have similar parity, demonstrating the efficacy of APGNN in learning task-specific graph filters.

**Polynomial Order $K$.** To gain insight into the role of polynomial order $K$, we conduct the experiment tuning $K$ in $\{1, 2, ..., 20\}$ on Cora, Citeseer, and Pubmed dataset. Our theoretical analysis supports the observation that a small $K$ can result in a large truncation error, leading to a low accuracy rate. It can be observed that the accuracy rate has little promotion when $K$ is larger than 10, although at the cost of high computational resources.

**Decay Rate $\alpha$.** Figure 3 (b) depicts the accuracy curve corresponding to various $\alpha$ values ranging from 0.1 to 0.9 and 0.99 on Cora, Citeseer and Pubmed datasets. As $\alpha$ decreases, the classification accuracy initially increases and then declines sharply. This phenomenon verifies the theory that the truncation error decreases as $\alpha$ decreases, but it leads to a trivial function when $\alpha$ is extremely small.

**$P$-hop strategy.** We investigate the accuracy associated with varying parameters $P$ taken from the set $\{1, 2, 3, 4, 5, 6\}$ when fixing $T = KP = 60$. As we can see in Figure 3 (c), the accuracy increase when $P > 1$. This phenomenon can be attributed to the fact that the generalization bounding decreases when $P$ increases, which suggests that the $P$-hop strategy can effectively explore deeper information with the same computational complexity.

## 6 Conclusion

This paper proposes a universal learning principle for a valid construction of GNN. An instantiation named APGNN is proposed to verify the effectiveness of our framework. APGNN employs a decay rate and a multiple $P$-hop strategy to learn the coefficients adaptively, which can efficiently aggregate the information from high-order neighbors. We present a theoretical analysis of the generalization capabilities of both our framework and APGNN, which provides a learning guarantee. Comprehensive experiments show the superior performance of APGNN. In the future, it is worth exploring diverse graph filters based on the proposed principle. As shown in the generalization analysis, the upper bound of the model complexity relies on $\mathcal{O}(\sqrt{\log K})$. How to devise the GNN with complexity free of the hyperparameter $K$ is also a meaningful research direction.

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
