| $57.79_{\pm0.11}$ | $61.20_{\pm0.08}$ | $73.23_{\pm0.05}$ | $65.66_{\pm0.20}$ | $87.79_{\pm0.42}$ |
| ChebNet | $79.92_{\pm0.18}$ | $70.90_{\pm0.37}$ | $76.98_{\pm0.16}$ | $63.24_{\pm1.43}$ | $90.76_{\pm0.73}$ |
| GCN | $82.03_{\pm0.27}$ | $71.05_{\pm0.33}$ | $79.26_{\pm0.18}$ | $72.05_{\pm0.45}$ | $92.07_{\pm0.13}$ |
| SGC | $81.89_{\pm0.26}$ | $\underline{72.18}_{\pm0.24}$ | $78.58_{\pm0.15}$ | $72.76_{\pm0.35}$ | $89.01_{\pm0.40}$ |
| GAT | $82.82_{\pm0.36}$ | $71.96_{\pm0.39}$ | $79.15_{\pm0.34}$ | $74.36_{\pm0.58}$ | $91.86_{\pm0.27}$ |
| GraphSage | $82.14_{\pm0.25}$ | $71.80_{\pm0.36}$ | $79.20_{\pm0.27}$ | $73.17_{\pm0.41}$ | $91.53_{\pm0.15}$ |
| PPNP | $83.73_{\pm0.31}$ | $71.74_{\pm0.44}$ | $80.28_{\pm0.22}$ | $74.69_{\pm0.53}$ | $92.58_{\pm0.06}$ |
| APPNP | $83.73_{\pm0.21}$ | $71.70_{\pm0.21}$ | $80.07_{\pm0.21}$ | $74.91_{\pm0.61}$ | $92.81_{\pm0.12}$ |
| GNN-LF(iter) | $\underline{83.83}_{\pm0.36}$ | $71.44_{\pm0.42}$ | $\underline{80.31}_{\pm0.16}$ | $75.19_{\pm0.49}$ | $92.78_{\pm0.22}$ |
| GNN-HF(iter) | $83.68_{\pm0.31}$ | $71.58_{\pm0.36}$ | $79.99_{\pm0.22}$ | $74.71_{\pm0.55}$ | $92.72_{\pm0.31}$ |
| DAGNN | $82.70_{\pm0.17}$ | $71.90_{\pm0.06}$ | $80.06_{\pm0.30}$ | $\underline{75.63}_{\pm0.48}$ | $\underline{93.24}_{\pm0.21}$ |
| Ours | $\mathbf{84.15}_{\pm0.23}$ | $\mathbf{72.44}_{\pm0.56}$ | $\mathbf{80.74}_{\pm0.24}$ | $\mathbf{76.03}_{\pm0.51}$ | $\mathbf{93.69}_{\pm0.20}$ |

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

## A Appendix

### A.1 Data statistics

Table 3: Data statistics for the node classification task.

| Dataset | Nodes | Edges | Features | Class | Train | Val | Test |
|---|---|---|---|---|---|---|---|
| cora | 2708 | 5429 | 1433 | 7 | 140 | 500 | 1000 |
| citeseer | 3327 | 4732 | 3703 | 6 | 120 | 500 | 1000 |
| pubmed | 19717 | 44338 | 500 | 3 | 60 | 500 | 1000 |
| wiki-cs | 11701 | 216123 | 300 | 10 | 200 | 500 | 1000 |
| ms academic | 18333 | 81894 | 6805 | 15 | 300 | 500 | 1000 |

### A.2 The detail of learned graph filters

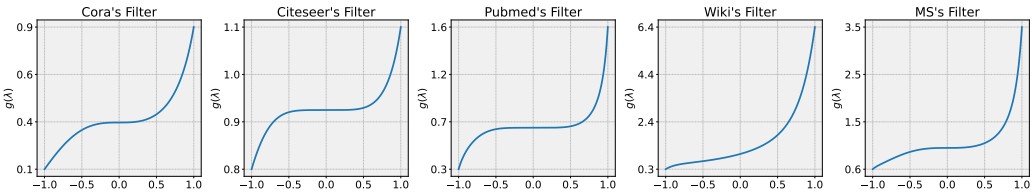

Figure 4: The graph filters learned using different data sets, with parameter $P$ being odd.

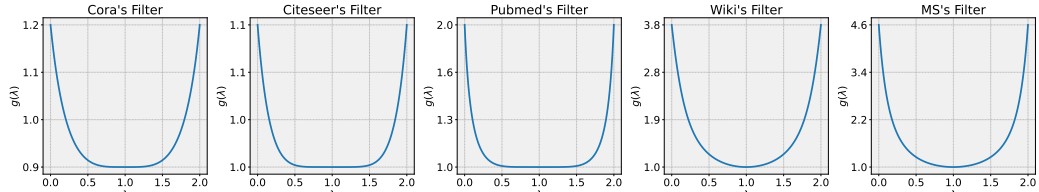

Figure 5: The graph filters learned using different data sets, with parameter $P$ being even.

### A.3 The hyperparameters settings

Table 4: The hyperparameters of APGNN on various datasets when parameter $P$ is odd.

| Dataset | $K$ | $P$ | $\alpha$ | Weight decay | Learning rate | Dropout rate |
|---|---|---|---|---|---|---|
| Cora | 10 | 3 | 0.7 | 0.005 | 0.01 | 0.8 |
| Citeseer | 10 | 5 | 0.7 | 0.00625 | 0.01 | 0.5 |
| Pubmed | 10 | 5 | 0.9 | 0.005 | 0.01 | 0.5 |
| Wiki-CS | 10 | 1 | 0.9 | 0.000525 | 0.03 | 0.4 |
| MS-Academic | 10 | 3 | 0.9 | 0.00525 | 0.02 | 0.4 |

### A.4 Proof for Lemma 1

($\Rightarrow$). We show the result by contradiction. If $\sum_k^\infty |a_k|$ is not convergent, then at $\gamma = 1$, we have $\sum_k^\infty |a_k \gamma^k|$ is not convergent, which occurs a contradiction. Therefore, series $\sum_k^\infty |a_k|$ converges.

($\Leftarrow$). It is obvious that for $\forall \gamma \in (-1, 1]$, $\sum_k^\infty |a_k \lambda^k| \leq \sum_k^\infty |a_k|$. Therefore, $\sum_k^\infty |a_k \lambda^k|$ uniformly converges in $\lambda \in (-1, 1]$. $\qquad \square$

Table 5: The hyperparameters of APGNN on various datasets when parameter $P$ is even.

| Dataset | $K$ | $P$ | $\alpha$ | Weight decay | Learning rate | Dropout rate |
|---|---|---|---|---|---|---|
| **Cora** | 10 | 4 | 0.7 | 0.005 | 0.01 | 0.8 |
| **Citeseer** | 10 | 6 | 0.7 | 0.00625 | 0.01 | 0.5 |
| **Pubmed** | 10 | 6 | 0.9 | 0.005 | 0.01 | 0.5 |
| **Wiki-CS** | 10 | 2 | 0.9 | 0.000525 | 0.03 | 0.4 |
| **MS-Academic** | 10 | 2 | 0.9 | 0.00525 | 0.02 | 0.4 |

## A.5 Proof for Theorem 1

$\tilde{\mathbf{A}}$ is an adjacency matrix of a graph, which is a real symmetric matrix. Since we can decompose $\tilde{\mathbf{A}}$ as $\tilde{\mathbf{A}} = \mathbf{U}\boldsymbol{\Gamma}\mathbf{U}^\top$, where $\mathbf{U}$ is a matrix composed of the eigenvectors of $\tilde{\mathbf{A}}$ and $\boldsymbol{\Gamma} = \text{diag}(\gamma_1, \cdots, \gamma_n)$ is the diagonal matrix of the corresponding eigenvalues. Therefore, we have

$$g(\mathbf{L}) = \sum_{k=1}^{\infty} \theta_k \tilde{\mathbf{A}}^k = \mathbf{U}\text{diag}\left(\sum_{k=1}^{\infty} \theta_k \gamma_1^k, \cdots, \sum_{k=1}^{\infty} \theta_k \gamma_n^k\right)\mathbf{U}^\top \tag{25}$$

Therefore, the $g(\mathbf{L})$ converges absolutely and uniformly if and only if $\sum_{k=1}^{\infty} \theta_k \gamma_i^k$ converges absolutely and uniformly for all $i \in [n]$. Then apply Lemma 1 and we can obtain the result. $\qquad\square$

## A.6 Proof of Theorem 2

We first introduce some definitions and Lemma for assisting with the proof.

**Definition 1.** Consider the sample set $S = \{\mathbf{x}_1, \cdots, \mathbf{x}_n\}$ and function set $\mathcal{F}$, where $f(\mathbf{x})$ is bounded for any $f \in \mathcal{F}$. Then the empirical Rademacher complexity is defined as:

$$\mathfrak{R}_S(\mathcal{F}) = \frac{1}{n}\mathbb{E}_{\boldsymbol{\sigma}}\left[\sup_{f \in \mathcal{F}} \sum_{i=1}^{n} \sigma_i f(\mathbf{x}_i)\right], \tag{26}$$

where $\sigma_i$ is i.i.d. Rademacher random variable defined by $\Pr(\sigma_i = -1) = \Pr(\sigma_i = 1) = 0.5$.

**Lemma 2.** *Consider the hypothesis set*

$$\mathcal{H}_{\mathcal{X}} = \{h : h(\mathbf{x}) = g_{\boldsymbol{\theta}} Lf(\mathbf{x}),\ f(\mathbf{x}) = \langle \mathbf{w}, \mathbf{x} \rangle,\ \|\mathbf{w}\|_2 \leq B,\ \|\boldsymbol{\theta}\|_1 \leq M\}, \tag{27}$$

*where $\boldsymbol{\theta} = [\theta_0, \theta_1, \cdots, \theta_K]$. Let $x_j$ denote the $j$-th element of $\mathbf{x} \in \mathcal{X}$, and $\mathbb{E}[x_j^2] \leq c_{\mathcal{X}}$ for any $j \in [d]$. Then for any sample set $S = \{\mathbf{x}_1, \cdots, \mathbf{x}_{n_l}\} \subset \mathcal{X}$ we have*

$$\mathfrak{R}_S(\mathcal{F}_{\mathcal{X}}) \lesssim 2BMc_{\mathcal{X}}\sqrt{\frac{2\log(2K+2)}{n_l}}. \tag{28}$$

*Proof.* Based on the definition, we can write

$$
\begin{aligned}
\mathfrak{R}_S(\mathcal{H}_{\mathcal{X}}) &= \frac{1}{n_l}\mathbb{E}_{\boldsymbol{\sigma}}\left[\sup_{h_{\mathbf{w},\boldsymbol{\theta}} \in \mathcal{H}_{\mathcal{X}}} \sum_{i=1}^{n_l} \sigma_i h_{\mathbf{w},\boldsymbol{\theta}}(\mathbf{x}_i)\right] \\
&= \frac{1}{n_l}\mathbb{E}_{\boldsymbol{\sigma}}\left[\sup_{\|\mathbf{w}\|_2 \leq B,\ \|\boldsymbol{\theta}\|_1 \leq M} \sum_{i=1}^{n_l} \sigma_i g_{\boldsymbol{\theta}} Lf(\mathbf{x}_i)\right] \\
&= \frac{1}{n_l}\mathbb{E}_{\boldsymbol{\sigma}}\left[\sup_{\|\mathbf{w}\|_2 \leq B,\ \|\boldsymbol{\theta}\|_1 \leq M} \sum_{i=1}^{n_l} \sigma_i \int_{\mathcal{X}} \sum_{k=0}^{K} \theta_k \tilde{A}^k(\mathbf{x}_i, \mathbf{x})\mathbf{x}^\top \mathbf{w} d\rho(\mathbf{x})\right] \\
&\leq \frac{B}{n_l}\mathbb{E}_{\boldsymbol{\sigma}}\left[\sup_{\|\boldsymbol{\theta}\|_1 \leq M} \left\|\sum_{i=1}^{n_l} \sigma_i \int_{\mathcal{X}} \sum_{k=0}^{K} \theta_k \tilde{A}^k(\mathbf{x}_i, \mathbf{x})\mathbf{x} d\rho(\mathbf{x})\right\|_2\right] \\
&= \frac{B}{n_l}\mathbb{E}_{\boldsymbol{\sigma}}\left[\sup_{\{\mathbf{v}_i\}_{i=1}^n \in V} \left\|\sum_{i=1}^{n_l} \sigma_i \mathbf{v}_i\right\|_2\right]
\end{aligned}
$$

where the inequality follows from the Cauchy-Schwarz inequality, and the set $V$ is defined as

$$V \triangleq \left\{ \{\mathbf{v}_i\}_{i=1}^n : \mathbf{v}_i = \int_{\mathcal{X}} \sum_{k=0}^K \theta_k \tilde{A}^k(\mathbf{x}_i, \mathbf{x})\mathbf{x}\mathrm{d}\rho(\mathbf{x}), \quad \|\boldsymbol{\theta}\|_1 \le M \right\}. \tag{29}$$

Define $q_j(\mathbf{x}) = x_j$ returning the $j$-th coordinate of the input. Hence, the $j$-th coordinate of $\mathbf{v}_i$ can be rewritten as

$$v_{ij} = \int_{\mathcal{X}} \sum_{k=0}^K \theta_k \tilde{A}^k(\mathbf{x}_i, \mathbf{x}) x_j \mathrm{d}\rho(\mathbf{x}) = \int_{\mathcal{X}} \sum_{k=0}^K \theta_k \tilde{A}^k(\mathbf{x}_i, \mathbf{x}) q_j(\mathbf{x}) \mathrm{d}\rho(\mathbf{x}) = \sum_{k=0}^K \theta_k \tilde{A}^k q_j(\mathbf{x}_i). \tag{30}$$

Since $\|\mathbf{u}\|_2 \le \sqrt{d}\|\mathbf{u}\|_\infty$ for any $\mathbf{u} \in \mathbb{R}^d$, we have

$$\mathfrak{R}_S(\mathcal{H}_{\mathcal{X}}) \le \frac{B\sqrt{d}}{n_l} \mathbb{E}_{\boldsymbol{\sigma}} \left[ \sup_{\{\mathbf{v}_i\}_{i=1}^n \in V} \left\| \sum_{i=1}^{n_l} \sigma_i \mathbf{v}_i \right\|_\infty \right]$$

$$\le \frac{B\sqrt{d}}{n_l} \mathbb{E}_{\boldsymbol{\sigma}} \left[ \sup_{\{\mathbf{v}_i\}_{i=1}^n \in V} \max_{j \in [d]} \left| \sum_{i=1}^{n_l} \sigma_i v_{ij} \right| \right]$$

$$\le \frac{2B\sqrt{d}}{n_l} \mathbb{E}_{\boldsymbol{\sigma}} \left[ \sup_{\{\mathbf{v}_i\}_{i=1}^n \in V} \max_{j \in [d]} \sum_{i=1}^{n_l} \sigma_i v_{ij} \right]$$

$$\le \frac{2B\sqrt{d}}{n_l} \mathbb{E}_{\boldsymbol{\sigma}} \left[ \sup_{\|\boldsymbol{\theta}\|_1 \le M} \sum_{i=1}^{n_l} \sigma_i \sum_{k=0}^K \theta_k \tilde{A}^k q_j(\mathbf{x}_i) \right]$$

$$= \frac{2B\sqrt{d}}{n_l} \mathbb{E}_{\boldsymbol{\sigma}} \left[ \sup_{\|\boldsymbol{\theta}\|_1 \le M} \sum_{k=0}^K \theta_k \sum_{i=1}^{n_l} \sigma_i \tilde{A}^k q_j(\mathbf{x}_i) \right]$$

$$= \frac{2BM\sqrt{d}}{n_l} \mathbb{E}_{\boldsymbol{\sigma}} \left[ \sup_{\boldsymbol{\theta} \in \Theta} \sum_{k=0}^K \theta_k \sum_{i=1}^{n_l} \sigma_i \tilde{A}^k q_j(\mathbf{x}_i) \right]$$

$$= 2BM\sqrt{d}\mathfrak{R}_S(\mathcal{H}'),$$

where $\Theta = \bigcup_{k=0}^K \{-\boldsymbol{e}_k, \boldsymbol{e}_k\}$ and $\boldsymbol{e}_k$ denote $k$-th vector with $k$-th entry as one and others are zero. The set $\mathcal{H}' = \{h(\mathbf{x}) = \sum_{k=0}^K \theta_k A^k q_j(\mathbf{x}) : \boldsymbol{\theta} \in \Theta\}$ is a finite set with $|\mathcal{H}'| = 2(K+1)$. We bound $\mathfrak{R}_S(\mathcal{H}')$ with Massart's Lemma:

**Lemma 3** (Massart's Lemma [21]). *Let $\mathcal{X} \subset \mathbb{R}^n$ be a finite set and $\sup_{\mathbf{x} \in \mathcal{X}} \|\mathbf{x}\|_2 \le r\sqrt{n}$, then the following inequality holds:*

$$\mathbb{E}_{\boldsymbol{\sigma}} \left[ \frac{1}{n} \langle \boldsymbol{\sigma}, \mathbf{x} \rangle \right] \le r\sqrt{\frac{2\log|\mathcal{X}|}{n}}, \tag{31}$$

*where $\boldsymbol{\sigma} = [\sigma_1, \cdots, \sigma_n]$ denote the vector of Rademacher random variables.*

Since $\mathcal{H}'$ is a finite set and for any $h \in \mathcal{H}'$,

$$\frac{1}{n_l} \sum_{i=1}^{n_l} h(\mathbf{x}_i)^2 = \frac{1}{n_l} \sum_{i=1}^{n_l} \sup_{k \in [n]} \left[ \tilde{A}^k q_j(\mathbf{x}_i) \right]^2$$

$$\approx \int_{\mathcal{X}} \sup_{k \in [n]} \left[ \tilde{A}^k q_j(\mathbf{x}) \right]^2 \mathrm{d}\rho(\mathbf{x}) \le \|q_j\|^2 \le c_{\mathcal{X}}^2.$$

where we use $\|\tilde{A}^k q_j\| \le \|\tilde{A}^k\|\|q_j\|$ and $\|\tilde{A}^k\| \le 1$ for any $k \in [n]$, and

$$\|q_j\|^2 = \int_{\mathcal{X}} q_j(\mathbf{x})^2 \mathrm{d}\rho(\mathbf{x}) = \int_{\mathcal{X}} x_j^2 \mathrm{d}\rho(\mathbf{x}) = \mathbb{E}[x_j^2] \le c_{\mathcal{X}}^2. \tag{32}$$

Therefore we finally obtain

$$\mathfrak{R}_S(\mathcal{F}_{\mathcal{X}}) \lesssim 2BMc_{\mathcal{X}} \sqrt{\frac{2d\log(2K+2)}{n_l}} \tag{33}$$

As a remark, we can present a more precise bound through McDiarmid's inequality. consider the convergence

$$\frac{1}{n_l} \sum_{i=1}^{n_l} \sup_{k \in [n]} \left[ \tilde{A}^k q_j(\mathbf{x}_i) \right]^2 \rightarrow \int_{\mathcal{X}} \sup_{k \in [n]} \left[ \tilde{A}^k q_j(\mathbf{x}) \right]^2 \mathrm{d}\rho(\mathbf{x}). \tag{34}$$

With the probability of at least $1 - \delta$,

$$\frac{1}{n_l} \sum_{i=1}^{n_l} \sup_{k \in [n]} \left[ \tilde{A}^k q_j(\mathbf{x}_i) \right]^2 \leq \int_{\mathcal{X}} \sup_{k \in [n]} \left[ \tilde{A}^k q_j(\mathbf{x}) \right]^2 \mathrm{d}\rho(\mathbf{x}) - \mathcal{O} \left( \sqrt{\frac{\log 1/\delta}{n_l}} \right). \tag{35}$$

The details are omitted since it is not the major part of the analysis. $\qquad\square$

*Proof.* We first write the excess risk decomposition:

$$R(h_{\mathbf{w},\boldsymbol{\theta}}) - \hat{R}(\hat{h}_{\mathbf{w},\boldsymbol{\theta}}) = \underbrace{R(h_{\mathbf{w},\boldsymbol{\theta}}) - \hat{R}(h_{\mathbf{w},\boldsymbol{\theta}})}_{A \text{ part}} + \underbrace{\hat{R}(h_{\mathbf{w},\boldsymbol{\theta}}) - \hat{R}(\hat{h}_{\mathbf{w},\boldsymbol{\theta}})}_{B \text{ part}} \tag{36}$$

For the $A$ part, we first apply Theorem 5.8 in [21]. With probability at least $1 - \delta$,

$$R(h_{\mathbf{w},\boldsymbol{\theta}}) - \hat{R}(h_{\mathbf{w},\boldsymbol{\theta}}) \leq \mathfrak{R}_S(\mathcal{H}_{\mathcal{X}}) + 3\sqrt{\frac{\log(2/\delta)}{2n_l}}. \tag{37}$$

Since the last term is of order $\mathcal{O}(\sqrt{\log(1/\delta)n_l^{-1}})$, which is significantly smaller than $\mathfrak{R}_S(\mathcal{H}_{\mathcal{X}})$, we rewrite the above inequality as

$$R(h_{\mathbf{w},\boldsymbol{\theta}}) \lesssim \hat{R}(h_{\mathbf{w},\boldsymbol{\theta}}) + 2BMc_{\mathcal{X}} \sqrt{\frac{2d \log(2K+2)}{n_l}}, \tag{38}$$

where we replace the Rademacher complexity with its upper bound by Lemma 2.

For the $B$ part, we first define the empirical operator over $S = \{\mathbf{x}_1, \cdots, \mathbf{x}_n\}$,

$$L_n f = \frac{1}{n} \sum_{i=1}^{n} \frac{A(\mathbf{x}_i, \cdot)}{\sqrt{d_n(\mathbf{x}_i) d_n(\cdot)}} f(\mathbf{x}_i), \quad d_n = \frac{1}{n} \sum_{i=1}^{n} A(\mathbf{x}_i, \cdot) \tag{39}$$

and $g_{\boldsymbol{\theta}} L_n = \sum_{k=0}^{K} \theta_k L_n^k$. Then we have

$$\begin{aligned}
\hat{R}(h_{\mathbf{w},\boldsymbol{\theta}}) - \hat{R}(\hat{h}_{\mathbf{w},\boldsymbol{\theta}}) &\leq \left| \hat{R}(h_{\mathbf{w},\boldsymbol{\theta}}) - \hat{R}(\hat{h}_{\mathbf{w},\boldsymbol{\theta}}) \right| \\
&\leq \frac{1}{n_l} \left| \sum_{i=1}^{n_l} h_{\mathbf{w},\boldsymbol{\theta}}(\mathbf{x}_i) - \hat{h}_{\mathbf{w},\boldsymbol{\theta}}(\mathbf{x}_i) \right| \\
&= \frac{1}{n_l} \left| \sum_{i=1}^{n_l} g_{\boldsymbol{\theta}} L f(\mathbf{x}_i) - g_{\boldsymbol{\theta}} L_n f(\mathbf{x}_i) \right| \\
&\leq \frac{1}{n_l} \left[ \sum_{i=1}^{n_l} (g_{\boldsymbol{\theta}} L f(\mathbf{x}_i) - g_{\boldsymbol{\theta}} L_n f(\mathbf{x}_i))^2 \right]^{1/2} \\
&\approx \| g_{\boldsymbol{\theta}} L f - g_{\boldsymbol{\theta}} L_n f \| \\
&\leq \| g_{\boldsymbol{\theta}} L - g_{\boldsymbol{\theta}} L_n \| \| f \|.
\end{aligned}$$

where the second inequality follows from the Lipschitz property. With Cauchy-Schwarz inequality,

$$\| f \|_2 = \int_{\mathcal{X}} \langle \mathbf{w}, \mathbf{x} \rangle \mathrm{d}\rho(\mathbf{x}) \leq B \cdot \mathbb{E}_{\mathbf{x}}[\|\mathbf{x}\|_2] \leq Bdc_{\mathcal{X}}. \tag{40}$$

According to Theorem 15 of [26], there exists a proper constant $C > 0$ related to $A(\cdot, \cdot)$, such that

$$\| L - L_n \| \leq \| L - L_n \|_{HS} \leq C \sqrt{\frac{\log(2/\delta)}{n}}. \tag{41}$$

464  with probability at least $1 - \delta$. Since the polynomial $g_{\boldsymbol{\theta}}$ is $L_M$-Lipschitz, we have

$$\|g_{\boldsymbol{\theta}} L - g_{\boldsymbol{\theta}} L_n\| \leq L_M C \sqrt{\frac{\log(2/\delta)}{n}} \tag{42}$$

465  Combining the above results, one can conclude that for any $(h_{\mathbf{w},\boldsymbol{\theta}}, \hat{h}_{\mathbf{w},\boldsymbol{\theta}}) \in \mathcal{H}_{\mathcal{X}} \times \mathcal{H}_S$,

$$R(h_{\mathbf{w},\boldsymbol{\theta}}) \lesssim \hat{R}(\hat{h}_{\mathbf{w},\boldsymbol{\theta}}) + 2BMc_{\mathcal{X}} \sqrt{\frac{2d \log(2K + 2)}{n_l}} + BCL_M dc_{\mathcal{X}} \sqrt{\frac{\log(2/\delta)}{n}}. \tag{43}$$

466  with probability at least $1 - \delta$. $\qquad\square$