# OpenReview forum: "Towards the Universal Learning Principle for Graph Neural Networks"
_NeurIPS.cc/2023/Conference — Submitted to NeurIPS 2023_

### Official Review · Reviewer_yz22 · 2023-06-27

**Soundness:** 2 fair
**Presentation:** 3 good
**Contribution:** 2 fair
**Rating:** 3
**Confidence:** 5

**Summary:**

The authors propose a spectral GNN with geometrically decaying weight $\alpha^k$ and $P$-hop polynomial basis. They study the Lipschitzness and generalization bound of the model. Experiment results demonstrate the good performance of the proposed method over baselines

**Strengths:**

(+) Interesting generalization analysis.

**Weaknesses:**

(-) The recent spectral GNN literature is not discussed or compared, where the advanced methods are also polynomial graph filters but with a more careful design (i.e., different basis).

(-) The motivation of the proposed new designs $\alpha$ and $P$-hop seems not convincing

(-) The spectral GNN baselines are completely missing in the experiment.

(-) The experiments do not demonstrate the benefit of the proposed design besides prediction accuracy. Since the authors talk about polynomial graph filters, I wonder how the proposed methods can learn different graph filters.

## Detail comments

I have mixed feelings about this paper. On one hand, I find the discussion on requiring the Lipschitz graph filter and generalization analysis interesting. On the other hand, I feel the authors did not conduct a comprehensive literature survey on the recent development of spectral GNNs, which makes me feel the paper is a bit outdated.

Note that the idea of learning polynomial graph filters has been extensively studied recently, as capable of learning high-pass graph filters enables good performance on heterophilic graph datasets [2]. Since GPR-GNN (using a monomial basis), many advances have been made. For instance, Bernnet [He et al. 2021] argue that using Bernstein polynomial basis has better properties. [Wang et al. 2022] further improve this line of work by using Jacobi polynomial basis. Both of them not only demonstrate great performance on homophilic and heterophilic datasets but also conduct experiments to directly examine how well spectral GNNs can learn each target graph filter such as low-pass, high-pass, band-pass and more. That is, these works show how to make $g_{\theta}$ learn better with different polynomial basis and thus constraints on the coefficients. I feel the proposed APGNN is very relevant to this literature and should be compared with them in both related works and experiments. Furthermore, Since APGNN is claimed to be a ”universal principle”, I wonder how well it can learn on heterophilic datasets and different graph filters (i.e., the experiment in [Wang et al. 2022] Table 1).

Also, I find the motivation for using $\alpha$ and $P$-hop not very convincing. For $\alpha$, I agree that if we set $K = \infty$ then the geometric decaying weight is needed. However, as the authors also mentioned in Section 4, we cannot use $K= \infty$ in practice. In this case, I wonder what benefit would $\alpha$ bring to us. Can not we just learn it automatically? On the other hand, the argument for motivating the use of $P$ seems problematic. Intuitively, we should compare two methods with the same number of hops ($K$). In this point of view, $P$ does not help in any sense. Also, if we fix $T$ (the maximum hop range), then the effect of increasing $P$ is merely reducing $K = T//P$, which apparently will make the generalization bound smaller. However, this is at the cost of increasing the term $\hat{R}$, otherwise one should simply set $K=0$.

Even from the experiment section, we can see that choosing $\alpha\approx 1$ gives roughly the same best-tuned results (Figure 3-(b)). A similar observation can be made for $P=1$ in Figure 3-(c), albeit the difference seems to be a bit larger. According to the argument above, I conjecture that the benefit of $P$ comes from having a smaller number of learnable weights (in the graph filter $g_{\theta}$). It is interesting to investigate deeper of how we can improve along this line but the current results and explanation are unsatisfactory. At last, I also hope the authors can repeat experiments of Figure 3 on heterophilic datasets and even on experiments of learning different graph filters (maybe too much to do though). I feel the finding therein can motivate the authors to some ideas to further improve the work.

Minor:

How can the authors answer "yes" in the reproducibility but not include their code (only after acceptance)? I feel the authors should be more serious about answering the checklist questions.


## References

[He et al. 2021] Bernnet: Learning arbitrary graph spectral filters via Bernstein approximation, He et al. NeurIPS 2021.

[Wang et al. 2022] How powerful are spectral graph neural networks, Wang et al. ICML 2022.


**Questions:**

1. Detail comparison with recent spectral GNN literature in both related works and experiments.

2. Conduct experiments on heterophilic datasets.

3. Conduct experiments in learning different graph filters.

4. Explain the benefit of $\alpha$ in finite $K$ regime.


**Limitations:**

I do not find potential negative social impact.

---

> ### Author Rebuttal · Authors · 2023-08-10
>
> Thanks for your comments and detailed suggestions to improve our work.
> In response to the points raised in your comments,
> we give the following discussion.
>
> 1.To explain the necessity of $\alpha$,
>     we look back to the motivation of the proposed learning framework.
>     Consider graph filter $g(\lambda)=\sum_{k=0}^K\theta_k(1-\lambda)^k$.
>     We observe the inconsistencies between GNNs and their infinite-depth versions, i.e., the graph filter cannot converge or possess good analytic property for $K\to\infty$.
>     This is the reason why some GNNs fail to perform better when increasing order $K$.
>     This motivates us to construct the framework that allows GNNs to keep consistency with the "infinite-depth" version.
>     Therefore, we propose the learning principle requiring convergence and the Lipschitz continuity of the graph filter.
>     From this point, we motivate $\alpha$, the exponential weight decay, following this principle.
>     The substantial purpose of $\alpha$ is to keep the "consistency".
>     This allows us to make $K\to\infty$,
>     and also allows us to apply large $K$ in practice.
>     On the other hand,
>     the proper $\alpha$ reduces the generalization risk by Proposition 1,
>     which claims the benefit of $\alpha$ in theory.
>     Besides, $\alpha$ is not a learnable parameter.
>     It is proposed as a hyperparameter to meet the learning principle.
>     Even if $\alpha$ is learnable,
>     it could cause exploding gradient during optimization.
>
> 2.We are thankful for your comprehensive advice for reviewing the related works.
>     We will include these works in the literature review and experiments.
>     Additional experiments are conducted to compare APGNN with spectral GNNs such as GPR-GNN and BernNet.
>     We also include the heterophilic datasets: Cornell, Wisconsin, and Texas.
>     The experimental results are shown in the attached PDF.
>     The results show that APGNN still outperforms GPR-GNN, BernNet, and other compared methods on most datasets (both homophilic and heterophilic datasets).
>     This suggests APGNN can learn more appropriate graph filters in the node classification tasks.
>     We also show the learned coefficients $\beta$ on different datasets.
>
> 3.The experiments of $P$-hop on heterophilic datasets are conducted and shown in the attached PDF.

---

> > ### Comment · Reviewer_yz22 · 2023-08-14
> >
> > I am not able to see the general response and the attached pdf that the authors mentioned. I keep my score unchanged.

---

### Official Review · Reviewer_UcBg · 2023-07-05

**Soundness:** 3 good
**Presentation:** 4 excellent
**Contribution:** 3 good
**Rating:** 6
**Confidence:** 4

**Summary:**

This paper theoretically studies the criterion for the graph filter formed by power series using Lipschitz smoothness and then proposes a novel Adaptive Power GN architecture. Some convergence and generalization analyses are covered. Experiments also show the advantages of the proposed methods, with some ablation studies on some parameters. Overall, it is a work that combines theory and practice. I think this work can benefit from some revisions. I am going to raise my score if my concerns are addressed. I would like to see a revision in the manuscript (it is OK to appear in the appendix) or a plan for the revisions (if the revision uploading is not allowed).

------------------------------------------------------------------------------------------------
After rebuttal, I increased my score to 6, which is based on my first impression and the responses. I am generally satisfied with the responses.

**Strengths:**

1. It is a paper that combines both theory and experiments.  I like the logic from theory to practice. Hence, the writing is excellent and clear to follow.
2. As a reviewer from the theoretical area, I think the generalization analysis on non-GCN graph neural networks is novel and exciting. One great part is that this analysis seems to cover a lot of existing graph neural networks. So it is general.
3. The experimental results support the theory and show the advantages.

**Weaknesses:**

1. Some parts of the paper are not very clear. It is mainly around lines 235 to 242. (a) For example, in line 239, it says, "larger $\alpha$ leads to a higher bound. " It is not clear what this "bound" refers to. At first, I thought it referred to $1-\alpha^K$ in line 235 or $1-\alpha^{T/P}$ in line 237. Later, I feel it should be the last two terms (or the second to the last term) in Equation 24. This needs clarification. (b) Another thing is at the end of this paragraph, it claims, "$\alpha$ should be moderate to...$. The discussion about small $\alpha$ is missing. I guess it is because a small $\alpha$ makes the filter trivial and not powerful (line 157). This also needs clarification.

2. Some discussions about theoretical works of generalization analysis on Graph Neural Networks are needed. I would like to know the comparison and theoretical novelties beyond existing works. Here are some recent related works.

[1] Esser et al., 2021, "Learning Theory Can (Sometimes) Explain Generalisation in Graph Neural Networks."

[2] Cong et al., 2021, "On Provable Benefits of Depth in Training Graph Convolutional Networks."

[3] Li et al., 2022, "Generalization guarantee of training graph convolutional networks with graph topology sampling."

[4] Zhang et al., 2023, "Joint Edge-Model Sparse Learning is Provably Efficient for Graph Neural Networks."

[5] Tang et al., 2023, "Towards Understanding Generalization of Graph Neural Networks."

**Questions:**

1. I think your theory covers other GNNs in Table 1. Can you use your Theorem 2 or another theory to show your proposed GNN is better than other GNNs in Table 1? It does not have to be very rigorous. I expect to see a comparison between different GNNs about $M$ and $L_M$ in equation 23.

2. I think the multiple P-hop message-passing strategy should work better on some heterophilous graphs. The citation graphs are homophilous. Could you please show some experimental results on heterophilous graph datasets?

**Limitations:**

There is no potential negative societal impact of this work.

---

> ### Author Rebuttal · Authors · 2023-08-09
>
> We really appreciate your high approval and constructive suggestions for our work!
> As you mentioned,
> this work provides a comprehensive framework with generalization analysis to pursue deeper and more effective GNN.
> Here we will address your concerns and state the revision plan.
>
> 1.Your understanding of the analysis around lines 235 to 242 (in "weakness 1") is correct.
>     The "bound" means the last two terms in equation (24),
>     which indicate the complexity and "quantization error" from continuous graph to discrete graph.
>     The reason why we do not use extremely small $\alpha$ is, as you said, to avoid the trivial filter.
>     From the spatial perspective, too small $\alpha$ also limits the information passing over the graph.
>     We plan to add detailed clarification in the revision.
>
> 2.We are thankful for your suggestion on the review of GNNs' generalization.
>     The related works are reviewed.
>     [1] and [5] present the generalization with transductive Rademacher complexity on node classification tasks (Note that their generalization error is only measured over the testing set).
>     In contrast, [2] analyze the transductive uniform stability of GNN (this is also related to [6]).
>     Considering the stochastic hypothesis, Ma et al. use PAC-Bayesian theorem to analyze the subgroup generalization bound of GNN [7].
>     [3] and [4] investigate the generalization guarantee of GNN via topology properties in the graph.
>     Different from these works,
>     we present the generalization guarantee over the whole sample space by extending the graph into its continuous version.
>     This is not restricted to the testing set and allows more general analysis with inductive learning.
>
> 3.Here we show some applications of Theorem 2.
>
> 1)PPNP: $M=1$, $L_M=\beta$, where $\beta>0$ is the hyperparameter of PPNP.
>
> 2)DAGNN: $M=K$, $L_M=O(K^2)$, where $K$ is the maximal order.
>         In this case, we can see that the complexity term of equation (24) becomes $O(K\log K)$.
>         Therefore, it tends to show weaker generalization compared with APGNN as $K$ increases.
>
> 3)GPR-GNN: $M=1$, $L_M=K$.
>
> In this case, the last two terms of RHS in (24) become $O(\log K)$ and $O(K)$.
>     We will add the above results as well as the analysis to the revision.
>
> 4.We conduct extra experiments on some heterophilic graph datasets, including Cornell, Texas, and Wisconsin.
>     These results will be appended to the revision.
>
> Additionally, we will give a further analysis of the tendency of $\beta$.
> The typos and some language mistakes will be corrected in the revision.
> We thank you again for the meaningful comments and suggestions,
> which inspire us to explore more interesting insights in this work.
>
> [1] Esser et al., 2021, "Learning Theory Can (Sometimes) Explain Generalisation in Graph Neural Networks."
>
> [2] Cong et al., 2021, "On Provable Benefits of Depth in Training Graph Convolutional Networks."
>
> [3] Li et al., 2022, "Generalization guarantee of training graph convolutional networks with graph topology sampling."
>
> [4] Zhang et al., 2023, "Joint Edge-Model Sparse Learning is Provably Efficient for Graph Neural Networks."
>
> [5] Tang et al., 2023, "Towards Understanding Generalization of Graph Neural Networks."
>
> [6] Verma et al., 2019, "Stability and generalization of graph convolutional neural networks."
>
> [7] Ma et al., 2021, "Subgroup generalization and fairness of graph neural networks."

---

> > ### Comment · Reviewer_UcBg · 2023-08-15
> >
> > Thank you for the response. My first impression of this paper is quite good. I expect a better performance on heterophilous graphs and hope it can improve this paper. I am generally satisfied with other answers to my questions. I have increased my score to 6, which is my initial evaluation.

---

### Official Review · Reviewer_8PH8 · 2023-07-06

**Soundness:** 2 fair
**Presentation:** 3 good
**Contribution:** 2 fair
**Rating:** 3
**Confidence:** 4

**Summary:**

This paper studies the polynomial filters in GNNs. Specfically, the authors propose a Adaptive Power GNN which employs exponentially decaying coefficients. A theoretically generalization  analysis of  the proposed framework is conducted. Experiments demostrate the proposed method can outperform some baselines on the selected datasets.

**Strengths:**

1. The background provided is detailed and aids readers in comprehending the nuances of the paper.
2. The authors offer a generalization analysis of the proposed method.
3. The paper is well-written and easy to follow.


**Weaknesses:**

1. The primary concern is the lack of novelty. The concept of polynomial filters in GNNs is well-established. Additionally, the paper looks at infinite orders, a domain where any function can be approximated by infinite polynomial functions. Also, the topic of GNN generalization has already been examined in several papers, such as [1], [2], and [3].
2. The method proposed in this paper comprises decoupled GNNs, which implies that the framework and analysis may not be applicable to coupled cases, such as GCN and GAT.
3. The hyperparamters $\alpha, \beta$ are hard to choose. For instance, GPR-GNN struggles to learn these hyperparameters effectively without proper initialization. How do the authors suggest these hyperparameters be selected?
4. The authors opted for fixed data splits across all datasets, which may predispose the model to overfitting on the valid/test sets. Have the authors considered random data splits? Additionally, the study could benefit from incorporating larger datasets like the OGB datasets.

[1] Baranwal, Aseem, Kimon Fountoulakis, and Aukosh Jagannath. "Graph convolution for semi-supervised classification: Improved linear separability and out-of-distribution generalization." arXiv preprint arXiv:2102.06966 (2021).

[2] Verma, Saurabh, and Zhi-Li Zhang. "Stability and generalization of graph convolutional neural networks." Proceedings of the 25th ACM SIGKDD International Conference on Knowledge Discovery & Data Mining. 2019.

[3] Ma, Jiaqi, Junwei Deng, and Qiaozhu Mei. "Subgroup generalization and fairness of graph neural networks." Advances in Neural Information Processing Systems 34 (2021): 1048-1061.


**Questions:**

Please refer to the weakness.

---

> ### Author Rebuttal · Authors · 2023-08-09
>
> We are thankful for your positive comment on our method.
> But it seems that the reviewer misunderstood some important facts about our work.
> We will further explain these points and dispel your concerns one by one.
>
> 1.It should be noted that the main topic of this paper is the design principle of graph filter consistent with infinite order.
>     To the best of our knowledge, this problem is not well investigated before.
>     There are numerous methods to explain the generalization of GNNs.
>     Though the existing works have given some ways to understand the generalization of GNNs,
>     the theoretical properties of GNNs are still not thoroughly studied.
>     In contrast, we explore the generalization with the learnable graph filter from the perspective of the continuous graph.
>     This provides a novel viewpoint to understand the GNNs' properties on inductive learning,
>     which is significantly different from the previous research,
>     including your listed references.
>     We still thank you for the advice on the review of GNNs' generalization.
>
> 2.The key to our work still lies in the design of graph filters.
>     The reason we use decoupled setting is the convenience for explanation and implementation.
>     In fact, our model is not contradictory to the coupled GNNs,
>     because we can treat it as one layer in coupled GNNs.
>     Therefore, we can easily extend our framework to coupled GNNs, like GPR-GNN [1].
>
> 3.To be clear, the parameter $\beta$ is the weight to be learned, not the hyperparameter.
>     Besides, The hyperparameter $\alpha$ is tuned via grid search in our experiments.
>     This is also shown in the experiment section.
>
> 4.The fixed data split is well-recognized by researchers and has been used in many works,
>     so we do not consider random split in our work.
>     We have conducted extra experiments on heterophilic datasets with the random split.
>     We will consider larger datasets in the extensive experiments.
>
> [1] Chien et al., 2021, "Adaptive universal generalized pagerank graph neural network."

---

> > ### Comment · Reviewer_8PH8 · 2023-08-18
> >
> > Thanks for the clarification from authors. I didn't see any further results. I tend to keep my score.

---

### Official Review · Reviewer_qgt5 · 2023-07-08

**Soundness:** 2 fair
**Presentation:** 1 poor
**Contribution:** 2 fair
**Rating:** 3
**Confidence:** 4

**Summary:**

In GNNs, designing a graph filter or propagation mechanism plays a critical role. Spectral-based GNNs formulate graph filters in the graph Fourier domain, and those filters are usually in the form of polynomials or power series. The manuscript argues that a well-defined graph filter should be convergent when represented as power series and have desirable analytic properties such as the Lipschitz continuity. Graph filters of four existing GNNs are analyzed based on the proposed conditions; it is shown that they have some limitations. The proposed APGNN introduces an exponentially decaying rate to coefficients of the learnable graph filter. Due to the decaying rate, the filter of APGNN is guaranteed to converge, and APGNN can theoretically be extended to an infinite-depth GNN. The filter of APGNN also satisfies the Lipschitz continuity, implying that the model is stable and robust. A mathematical analysis of the generalization bound of APGNN is provided. In practice, a truncated polynomial instead of the power series is utilized as the filter of APGNN to avoid the infinite number of learnable parameters. A multiple P-hop strategy, which uses the P-th power of the adjacency matrix, is introduced to enlarge the receptive field of the graph filter while maintaining the same computational complexity. Experimental results on five real-world benchmark datasets show that APGNN has comparable performance to 11 baselines in accuracy.

**Strengths:**

1. The manuscript points out that there exist inconsistencies between GNN models and their infinite-depth versions. A criterion for the polynomial-based graph filter is proposed with two constraints: the convergence and the Lipschitz continuity of the graph filter. The criterion reflects the stability of a GNN model with respect to the input graph and the consistency between a GNN and its infinite-depth version.

2. The motivation of APGNN is well described. The criterion for a polynomial-based graph filter is proposed, and the formulation of APGNN is provided as an instantiation of a graph filter satisfying the criterion. A truncated polynomial filter is suggested for practical use where the number of parameters should be finite.

3. The effectiveness of the P-hop strategy is shown in experiments. In Figure 3 (c), the P-hop strategy allows the model to keep the performance while decreasing the computational cost by omitting the terms that are not multiple of P from the graph filter polynomial.

**Weaknesses:**

1. There should be a comparison between APGNN and other GNN methods with an infinite depth. Existing approaches utilize a residual connection [1] or formulate the state of equilibrium [2, 3] to model GNNs with an infinite depth. Those methods should be theoretically and empirically compared to APGNN.

   [1] Chen et al., Simple and Deep Graph Convolutional Networks, ICML 2020.

   [2] Gu et al., Implicit Graph Neural Networks, NeurIPS 2020.

   [3] Liu et al., EIGNN: Efficient Infinite-Depth Graph Neural Networks, NeurIPS 2021.

2. The node classification accuracy of APGNN is relatively low compared to the existing GNN models such as GCNII [1] and G$^2$CN [4]. For example, the accuracy of GCNII is 85.5 on Cora, whereas the accuracy of G$^2$CN is 73.8 on Citeseer. In addition, GNN models with an infinite depth should be compared to APGNN.

   [4] Li et al., G$^2$CN: Graph Gaussian Convolution Networks with Concentrated Graph Filters, ICML 2022.

3. The authors mentioned the hyperparameter sensitivity of PPNP in lines 130-131: "However, the performance of PPNP is heavily dependent on the hyperparameter $\beta$, which must be carefully tuned to achieve optimal performance." However, the same applies to APGNN since the node classification performance heavily depends on $\alpha$, a decay weight. Figure 3(b) shows that the accuracy gap between the best and the worst cases is more than 20% on Cora.

4. The polynomial order K is fixed to 10 for some baseline methods, such as ChebNet and DAGNN. However, the optimal value of K can vary depending on the baseline methods. In [1], the node classification results with various depths (Table 3 of [1]) show that the optimal depths are different for each combination of model and dataset. Therefore, the authors should show the node classification results with various polynomial orders, as in [1], or tune the polynomial order for each baseline method, as in [5].

   [5] Liu et al., Towards Deeper Graph Neural Networks, KDD 2020.

5. Minor Comments
- In line 54, $i = [n_l]$ should be modified to $i \in [n_l]$.
- There is an inconsistency between Equations (2) and (3).
- In line 93, "L-Lipschtiz" should be changed to "L-Lipschitz".
- In line 136, “DAGGN” should be modified to “DAGNN”
- In Equation 13, $(-1)^k$ should be modified to $(-1)$.
- $\rho$ indicates the spectral radius in Theorem 1, whereas it represents a probability measure in Section 4.
- In Lines 192-195, there is no description and definition of $c_{\mathcal{X}}$, $c_{\mathcal{U}}$, $c_{\mathcal{L}}$.
- In line 206, ‘the hypothesis set over __ is described as’ should be modified to ‘the hypothesis set over \mathcal{X} is described as’.
- In line 266, APPNP is mentioned twice.

**Questions:**

1. Is there any tendency on the learned coefficients of the graph filter $\beta$? APGNN learns the coefficients $\beta$ during training to deal with arbitrary filter shapes. A higher $\beta$ value means that the corresponding hop of neighbors is more important than a hop with a lower $\beta$ value. Thus, by analyzing the learned $\beta$, more explanations about the neighborhood depth can be added, such as the appropriate polynomial order for the model or important hops of the dataset.

2. What is the total runtime of APGNN concerning K? As K grows, the proposed graph filter will become denser since its receptive field becomes larger. This aspect can increase the computation cost of applying the proposed graph filter. Meanwhile, the graph filter of GCN is relatively sparse since it sets K to 1. Therefore, comparing the runtimes of GNN models as K increases is encouraged.

**Limitations:**

1. There might be some cases where capturing the long-range dependencies is essential to understand the context of the given graph [6]. However, the proposed model might not be appropriate for those graphs. To capture long-range dependencies, either the decay weight $\alpha$ has to be large enough, or $\beta$ assigned to the low-order should be small. The former increases the generalization bound of the model, and the latter might lead to a vanishing gradient problem.

   [6] Dwivedi et al., Long Range Graph Benchmark, NeurIPS 2022.

2. The performance of the model is highly sensitive to the choice of $\alpha$.

---

> ### Author Rebuttal · Authors · 2023-08-09
>
> We are thankful for your detailed and constructive suggestions. We carefully consider your meaningful questions and hope the following elaboration can answer your concern.
>
> 1.We discuss the essential difference of "infinite depth" in the mentioned works.
>     GCNII stacks multiple graph convolutional layers elaborated with the residual connection and identity mapping.
>     The author claims that in theory, the number of stacking layers can go infinite.
>     IGNN and EIGNN share a similar idea of finding the state of equilibrium.
>     The "infinity" behind these methods refers to the stationary point (fixed point), or the state that further propagation does not change the data distribution.
>     In our work,
>     we proposed the framework to design the graph filter formed with power series (infinite order polynomial).
>     The convergence and Lipschitz properties are required.
>     This leads to the principle for constructing GNN consistent with its "infinite" version.
>     APGNN is designed following this principle and thus its order $K$ can go infinite (at least theoretically).
>     Though $K$ is set as a finite number in practice,
>     due to the expressive ability of the power series,
>     APGNN can approximate any appropriate filter in controllable precision by learning (equation (14)).
>     The experimental results of GCNII, IGNN, and EIGNN are shown in the attached PDF.
>
> 2.We have added the experiments comparing APGNN with more related methods.
>
> 3.We cannot compare the sensitivity of different methods by setting their hyperparameters as extreme values.
>     This is because the hyperparameters should work in a rational range.
>     In fact, we have discussed the property of $\alpha$ in the APGNN in section 3.3 and section 4.
>     The extremely small $\alpha$ tends to induce a trivial graph filter.
>     Therefore, the small $\alpha$ is not recommended in practice.
>     In our experiments,
>     the performance of APGNN is relatively stable when $\alpha$ changes in $[0.6, 0.99]$ in most datasets.
>     These results not only actually suggest the robustness of $\alpha$,
>     but demonstrate the reasonable range of $\alpha$.
>     Therefore, the accuracy gap cannot declare the sensitivity of hyperparameters.
>     In contrast, the performance of PPNP varies more than 20$\%$ on Cora when $\alpha$ changes in $[0.6, 0.99]$.
>     We also see similar phenomena in other datasets during our empirical study.
>     i.e., the performance of PPNP might drop rapidly with the slight change of $\alpha$.
>     Therefore, we say the performance of PPNP heavily depends on the hyperparameter.
>
> 4.In the early experiments,
>     we tuned the maximal order $K$ for ChebNet and DAGNN.
>     We observed that there is no significant performance difference between $K=10$ and larger $K$.
>     Thus, we directly set $K=10$ for these methods.
>
> 5.Thanks for the advice for discussion on the tendency of the learned coefficients $\beta$.
>     The experimental results show that
>     the sign of learned coefficients $\beta_i$ tends to be the same for all $i\in[K]$ on homogeneous datasets such as Cora, Citeseer, and Pubmed.
>     But for the heterophilic datasets like Cornell, Wisconsin, and Texas,
>     we usually obtain $\beta$ with both positive and negative elements.
>     The reason might be that feature propagation on heterogeneous graphs is not always beneficial.
>     Thus, the propagation of some hops should reach the opposite sign to suppress the inappropriate propagation.
>
> 6.We conducted extra experiments on the time comparison with different $K$.
>     APGNN costs less time during the training process than other methods.
>     The time tends to grow linearly as $K$ increases.
>     The experimental results are shown in the attached PDF.
>
> 7.Long-range dependence is not part of our study. However, we can still capture such dependence via setting relatively large $\alpha$ (e.g. 0.95).
>
> 8.We really appreciate your careful review of our manuscript.
>     The typos and mistakes will be corrected in our future revision.

---

> > ### Comment · Reviewer_qgt5 · 2023-08-17
> >
> > I cannot find the attached PDF the authors mentioned. Since I cannot check the results, I want to keep my score.

---

### Official Review · Reviewer_3RkM · 2023-07-08

**Soundness:** 3 good
**Presentation:** 3 good
**Contribution:** 3 good
**Rating:** 5
**Confidence:** 4

**Summary:**

The paper proposes a regularized learning framework for creating deep Graph Neural Networks (GNNs), including the Adaptive Power GNN (APGNN) that uses exponentially decaying weights to aggregate graph information of varying orders. The proposed multiple P-hop message passing strategy efficiently perceives higher-order neighborhoods, and the APGNN can be extended to an infinite-depth network.

**Strengths:**

1. The proposed APGNN model effectively captures higher-order neighborhood information, which is a crucial aspect of many graph-based tasks. This is a significant contribution to the field.
2. The regularized learning framework utilized in the proposed model provides a valuable theoretical guarantee for convergence. This adds to the credibility and reliability of the approach.
3. The experimental results presented in the paper demonstrate that the proposed method outperforms other state-of-the-art GNN models on several benchmark datasets.

**Weaknesses:**

1. It would be beneficial for the paper to include a more detailed comparison with other recent GNN models that also aim to capture higher-order neighborhood information. This would provide readers with a clearer understanding of how the proposed APGNN model compares in terms of performance and capabilities.
2. The paper lacks a detailed analysis of the computational complexity of the proposed method. Considering the potential concerns related to large-scale graphs, it is important to provide insights into the computational requirements of the model.
3. Similarly, the paper should include a more detailed analysis of the sensitivity of the proposed method to hyperparameters. Understanding how the model's performance varies with different hyperparameter settings is crucial for practical implementation.
4. The paper lacks a thorough analysis of the interpretability of the learned graph filters. Providing insights into the interpretability of the model's learned representations would enhance the understanding and trustworthiness of the proposed approach.


**Questions:**

1. Can you provide more details on the computational complexity of the proposed method, and how it scales with the size of the graph?
2. How sensitive is the proposed method to the choice of hyperparameters, and how did you select the hyperparameters used in the experiments?
3. Can you provide more details on the interpretability of the learned graph filters, and how they can be used to gain insights into the structure of the graph?
4. Have you considered the robustness of the proposed method to adversarial attacks, and how it compares to other GNN models in this regard?
5. How do you plan to extend the proposed method to handle dynamic graphs, where the structure of the graph changes over time?

**Limitations:**

Considering the increasing importance of robustness in real-world applications, it would be valuable to evaluate the model's performance under adversarial scenarios and discuss its limitations and strengths in this regard.

---

> ### Author Rebuttal · Authors · 2023-08-10
>
> We are grateful for your positive comments and meaningful suggestions.
> We hope the following explanation can help you better understand our work.
>
> 1.Here we give the analysis of complexity.
>     Denote $N$ as the number of nodes,
>     $E$ as the set of edges,
>     $d$ as the hidden layer of MLP,
>     and $c$ as the number of classes.
>     The adjacency matrix is stored in sparse format (i.e., only the edge will be stored),
>     so the space complexity is $O(|E|)$.
>     In our implementation, we use a 2-layer MLP for feature extraction and $K$-order polynomial graph filter.
>     Therefore, the time complexity of MLP and graph convolution is $O(NdcL)$ and $O(Kc|E|)$, respectively.
>     Therefore, the complexity grows linearly as the number of samples increases, and it also depends on the number of edges in the graph.
>
> 2.As shown in Fig. 3,
>     the performance is relatively steady when the hyperparameters vary within a reasonable range.
>     As for the selection of hyperparameters, the grid search method is used.
>
> 3.From Fig. 1, we can observe that the sign of learned coefficients
>  tends to be the same for all
>  on homogeneous datasets such as Cora, Citeseer, and Pubmed. But for the heterophilic datasets like Cornell, Wisconsin, and Texas, we usually obtain with both positive and negative elements. The reason might be that feature propagation on heterogeneous graphs is not always beneficial. Thus, the propagation of some hops should reach the opposite sign to suppress the inappropriate propagation. The tendency of the coefficient can further reflect the homophilic and heterogeneity of the graph.
>
> 4.The core contribution of this paper is the establishment of the principle of graph filters.
>     The related model design and the theoretical properties are more significant in our work.
>     Therefore, it does not involve discussion on the robustness of adversarial attacks and dynamic graphs.

---

### Decision · Program_Chairs · 2023-09-21

**Decision:**

Reject

**Comment:**

The paper introduces a regularized learning framework designed for the construction of deep Graph Neural Networks (GNNs). This framework includes the Adaptive Power GNN (APGNN), which employs exponentially decaying weights to effectively aggregate graph information across different orders. The approach incorporates a multi-P-hop message passing strategy that enables the efficient exploration of higher-order neighborhoods. Furthermore, the APGNN can be extended to create GNNs with indefinite depth.

The proposed method may have some merits.  However, one of the concerns of the reviewers is the difference from the existing methods, where there exist many related work, and it is still not fully addressed during the rebuttal. To address the concern, it would needs more comparison with existing methods. Thus, this paper is not ready for publication. I encourage authors to revise the paper based on the reviewer's comments and resubmit it to a future venue.